

# Assimilation of volcanic sulfur dioxide products from IASI and TROPOMI into the chemical transport model MOCAGE: case study of the 2021 La Soufrière Saint-Vincent eruption

Mickaël Bacles [1], Jonathan Améric [1], and Vincent Guidard [1]

[1]CNRM, Université de Toulouse, Meteo France, CNRS, Toulouse, France

**Correspondence:** Mickaël Bacles  (mickael.bacles@meteo.fr)

**Abstract.** Sulfur dioxide emitted during volcanic eruptions can be hazardous for aviation safety. As part of their activities, the Volcanic Ash Advisory Centres (VAACs) are therefore interested in the real-time atmospheric monitoring of this gas. A recent development aims at improving the forecasts of volcanic sulfur dioxide quantities made by the MOCAGE chemistry transport model. For this purpose, observations from both TROPOMI and IASI (B and C) instruments located on separate polar orbiting satellites are assimilated in the model. These sulfur dioxide measurements are based on the eruption event of the La Soufrière Saint-Vincent volcano in April 2021. Observations from the OMI instrument are considered as validation data. The resulting assimilation experiments show that the combined assimilation of IASI and TROPOMI observations always leads to a better forecast compared to the independent assimilation of data from each instrument. Sulfur dioxide atmospheric field forecasts are better when the available observations are numerous and cover a long time window.

## 1 Introduction

During volcanic eruptive events, large quantities of ash and sulfur dioxide ($SO_2$) are quickly released into the atmosphere. The emitted volcanic plumes can be transported far from the emission sources, reaching sometimes the upper troposphere or even the stratosphere (Carn et al., 2009). At such altitudes, volcanic ash plumes become hazardous for aviation safety as they can irreversibly damage aircraft engines and significantly lower flight visibility (Prata, 2009). Aircraft passengers and crew are also directly threatened, especially because air quality inside and at the vicinity of volcanic plumes is strongly degraded, generating respiratory issues detrimental to human health (Schmidt et al., 2014).

Some volcanoes are monitored by in situ sensors except if they are hardly reachable and hazardous. Consequently, passive satellite remote sensing remains an efficient technique providing global data on gases and aerosols emitted during volcanic eruptions. Sulfur dioxide is one of the compounds measurable by remote sensing. The absorbing bands of this gas are in the ultraviolet (UV $\sim 310 - 340$nm) and thermal infrared (IR - $\nu_1 \sim 8.6\mu$m, $\nu_3 \sim 7.3\mu$m and $\nu_1 + \nu_3 \sim 4\mu$m) domains (Carn et al., 2016).

In many eruption cases like the one of the Icelandic volcano Eyjafjallajökull in 2010, volcanic sulfur dioxide can be considered as a tracer to predict volcanic ash dispersion (Sears et al., 2013). However, both ash and sulfur dioxide plumes need to be monitored separately as their spatial distribution do not always coincide perfectly (Thomas and Prata, 2011). A striking



example is the eruption case of the Icelandic volcano Grímsvötn in 2011, during which both sulfur dioxide and ash plumes
were clearly separated (Prata et al., 2017).

Volcanic sulfur dioxide primary emissions can also be rapidly converted into secondary sulfate aerosols by reacting with
water vapour and dioxygen. This conversion directly impacts the spatial distribution of volcanic sulfur dioxide plumes. The
eruption of the Hunga Tonga volcano in January 2022 was exceptional as volcanic gases (sulfur dioxide and water vapour)

and ash have been injected at least at 30km altitude (Witze, 2022). As a result, stratospheric sulfur dioxide has been rapidly
converted into sulfate aerosols because of water vapour propelled in the stratosphere during the submarine eruption (Sellitto
et al., 2022).

To guarantee aviation safety, pilots need to dispose of accurate data on volcanic plume extent, movement and chemical com-
position. The International Civil Aviation Organization (ICAO) and the World Meteorological Organization (WMO) created in

1987 the International Airways Volcano Watch (IAVW) for this purpose (Lechner et al., 2018). Since 1990, the IAVW system
includes nine worldwide responsibility areas, each represented by a Volcanic Ash Advisory Centre (VAAC). Europe, Africa
and Middle East are part of the Toulouse VAAC supervised by Météo-France (Gouhier et al., 2020).

Information on volcanic plumes provided by each VAAC to aviation authorities currently rely on specific services like the
Support to Aviation Control Service (SACS) (Brenot et al., 2014) or the European Natural Airborne Disaster Information and

Coordination System for Aviation (EUNADICS-AV) (Brenot et al., 2021). Based on in-situ measurements, satellite data and
modelling products, these systems provide to the VAACs information on volcanic ash and sulfur dioxide plumes. The Toulouse
VAAC forecasts the dispersion of volcanic ash plumes by running MOCAGE-Accident (Gouhier et al., 2020), a specific version
of the three-dimensional chemistry transport model (CTM) MOCAGE (Modèle de Chimie Atmosphérique à Grande Échelle)
of Météo-France. To achieve this, an injection profile and a quantity of ash emitted by the volcano, previously chosen by the

forecaster, are used to predict the dispersion of the volcanic plume.

Assimilation of volcanic sulfur dioxide observations into a model has already been performed in several situations like
for the eruption events of the Eyjafjallajökull in 2010 and the Grímsvötn in 2011. Volcanic sulfur dioxide released by these
volcanoes has been monitored by the GOME-2 (Global Ozone Monitoring Experiment-2) and OMI (Ozone Monitoring Instru-
ment) UV sensors. The resulting retrieved observations have been assimilated in the Integrated Forecasting System (IFS) of

the European Centre for Medium-Range Weather Forecasts (ECMWF) by using the former Monitoring Atmospheric Compo-
sition and Climate (MACC) system dedicated to atmospheric composition. Initialising the model with sulfur dioxide analysis
improved the sulfur dioxide plume forecasts (Flemming and Inness, 2013). Volcanic sulfur dioxide retrievals from GOME-2
and TROPOMI (Tropospheric Monitoring Instrument) UV sensors observations have been assimilated in the global CAMS
(Copernicus Atmosphere Monitoring Service) assimilation system after the 2019 Raikoke volcanic eruption. On top of that, a

retrieved volcanic sulfur dioxide layer height from the TROPOMI Layer Height product has been assimilated. Including plume
height information into the assimilation system enhanced the quality of the forecasts made from the analyse fields (Inness et al.,
2022).

In April 2021, the eruption of the La Soufrière Saint-Vincent volcano emitted a large amount of $SO_2$ into the atmosphere.
This sulfur dioxide has been detected by several remote sensing instruments like the IR sensor IASI (Infrared Atmospheric



Sounding Interferometer), the UV sensors TROPOMI and OMI. The assimilation of TROPOMI, IASI and both instruments should improve MOCAGE. Indeed, without data assimilation, the model does not simulate $SO_2$ plume. In this study, we assimilate jointly an UV instrument, TROPOMI, and IR instruments, IASI B and IASI C allowing to correct the model during both day and night. The use of instruments using different wavelengths to assimilate volcanic SO2 data is also beginning to be developed in IFS. In MOCAGE, a part of the $SO_2$ can be converted into sulfate aerosols, particles which are causing problems in the aviation sector.

For the assimilation, the three-dimensional variational data assimilation system of the CTM MOCAGE (El Amraoui et al., 2022) is used. Forecasts of volcanic sulfur dioxide plumes are then initialized by the resulting analyses. Two preliminary experiments are conducted by assimilating independently total columns retrievals from IASI and TROPOMI sensors. The resulting total columns of volcanic sulfur dioxide assimilated in MOCAGE are then compared to those measured by the OMI independent sensor, located on another satellite.

In this paper, the La Soufrière Saint-Vincent eruption event of 2021 is introduced in the second part. Then, the assimilation data provided by the corresponding instruments is described in the third part, before the chemical transport model MOCAGE and its assimilation system in the fourth part. The fifth parts address the results of this case study.

## 2 Description of the 2021 La Soufrière Saint-Vincent eruption

La Soufrière Saint-Vincent is a volcano located in the Grenadines islands (13.33°N, 61.18°W). The eruption in 2021 started on 9[th] April with a violent explosion around 12:40 UTC local time. This first explosion released a volcanic plume that reached an altitude of 8 km. As a result, thousands of people were forced to flee. A second and weaker explosion occurred at 18:45 UTC, generating a volcanic plume that reached an altitude of 4 km. At 22:35 UTC, a third explosion took place with a plume reaching 16 km. Between 10[th] April and 11[th] April in the morning, the volcanic activity became periodic as explosions occurred every 1 to 3 hours, during short time periods of 20 to 30 minutes each. Although the number of explosions decreased from 12[th] April, the volcanic plume remained at a high altitude, exceeding 12 km and sometimes even 16 km. Two last major explosions took place on 12[th] April at 08:15 UTC and on 13[th] April at 10:30 UTC with plumes reaching altitudes of 12.8 km and 11 km respectively. The volcano continued to emit temporarily ash and volcanic gases in the atmosphere until the 22[nd] April but the plume from these explosions did not reach 8km high anymore. No fewer than 30 explosions were observed during this eruption event, most of them during 9[th] April and 11[th] April. More information about the La Soufrière eruptions is available in the report of Bennis and Venzke (2021).

## 3 Description of the instruments

### 3.1 TROPOMI

TROPOMI is a hyperspectral radiometer with spectral bands extending from the UV to the shortwave infrared (SWIR) domains. This instrument is on board the polar orbiting Sentinel 5 Precursor (S5p) satellite whose goal is to provide information and



services on air quality and climate (European Space Agency, 2020). Since August 2019, TROPOMI benefits from a high spatial resolution of 5.5 km x 3.5 km at nadir. This instrument has a daily temporal resolution (no observation at night) and its overpass local time occurs at 13:35 UTC (Veefkind et al., 2012).

Volcanic sulfur dioxide plumes can be globally monitored with the high spatial resolution of TROPOMI. Unprocessed radiances measured by the instrument are often unpacked and formatted to become level 1 (L1) data. This data can then be converted into a proper retrieved environmental variable as sulfur dioxide, forming a level 2 (L2) product. This conversion requires the use of a retrieval algorithm having its own specificities and uncertainties. Historically, the first institutes elaborating sulfur dioxide L2 products with TROPOMI measurements are the Royal Belgian Institute for Space Aeronomy (BIRA) and the German Aerospace Center called DLR (Deutsches Zentrum für Luft- und Raumfahrt). For that, they use the differential optical

absorption spectroscopy (DOAS) algorithm, particularly fitted for TROPOMI operational near-real time processing (European Space Agency, 2020).

Backscattered ultraviolet radiation is measured in order to construct absorption spectra. DOAS algorithm is then applied to these spectra for different fitting windows between 310 nm and 390 nm. The DOAS algorithm operates in several steps. First, slant column densities (SCD) are computed. They correspond to the integrated sulfur dioxide concentration along the mean

atmospheric optical path. Then, conversion factors called Air Mass Factors (AMF) are obtained from suitable radiative transfer calculations to take measurement sensitivity changes into account. These changes depend on many factors like observation geometry, total ozone absorption, clouds, surface reflectivity. Moreover, measurement sensitivity varies with the altitude of the emitted sulfur dioxide plume. As this altitude is unknown, the AMF are computed for several hypothetical sulfur dioxide vertical profiles. One profile used for polluted scenarios comes from a forecast made by the TM5 chemical transport model

(Huijnen et al., 2010). Three other profiles are available for 1 km thick boxes. The first box extends from the ground level to 1 km high. The two others are centred at 7 and 15 km above mean sea level. The first profile is located in the boundary layer and stands for well-mixed anthropogenic or volcanic sulfur dioxide conditions. The second profile aims at representing sulfur dioxide emitted by effusive volcanic eruptions in the upper troposphere. The third one is for sulfur dioxide released by explosive volcanic eruptions above the lower stratosphere (Theys et al., 2017). As four AMF are available depending on

different assumed $SO_2$ vertical profiles, the conversion of SCD into vertical column densities (VCD) generates four types of VCD. These vertical columns correspond to the number of sulfur dioxide molecules in an atmospheric column per unit area, usually expressed in Dobson Unit (1 DU = $2.69 \cdot 10^{16}$ molecules.cm$^{-2}$). Finally, averaging kernels are computed for the four vertical profiles (Theys et al., 2019).

For our study, we use sulfur dioxide total vertical columns computed from the hypothetical profile centred around 15 km.

These columns are associated to their systematic errors in order to compute the observation error matrix of the MOCAGE assimilation system. Moreover, the averaging kernel matrix needs to be considered for comparing TROPOMI to other instrumental measurements or model calculations (Rodgers and Connor, 2003). Finally, data is selected according to the category of the TROPOMI detection. Many cases are taken into consideration in our study: flag 1 for sulfur dioxide detection, 2 for clear volcanic detection and 3 for detection close to a known anthropogenic source. The quality of the $SO_2$ retrieval is given by a



quality flag with values ranging from 0 for uncertain retrieval to 1 for the best retrieval. TROPOMI data are available on the NASA website (https://disc.gsfc.nasa.gov/, last access: 19[th] September 2024).

## 3.2 TROPOMI Layer Height product

The TROPOMI Layer Height product (Hedelt et al., 2019) allows to know the altitude of a $SO_2$ plume when TROPOMI $SO_2$ total columns are higher than 20 DU thanks to a machine learning algorithm called "Full-Physics Inverse Learning Machine"

(FP_ILM). Hedelt et al used the LInearized Discrete Ordinate Radiative Transfer model (LIDORT) (Spurr et al., 2008) to simulate many reflectance spectra for different values of solar zenith angle (SZA), viewing zenith angle (VZA), relative azimuth angle (RAA), $O_3$ and $SO_2$ vertical column density, layer height, surface albedo and surface pressure. Before classifying these reflectance spectra in ten principal components (PC), TROPOMI instrument spectral response function, characterising the sensitivity of the instrument across its measurement spectrum, is applied. These PCs and information about the surface,

$O_3$ total columns, SZA, VZA and RAA are used as an input of the neural network. $SO_2$ total columns and $SO_2$ height are diagnosed thanks to the neural network. In this study, $SO_2$ total columns diagnosed by the neural network are not assimilated. Nevertheless, $SO_2$ height product is used to validate the altitude of the modelled plume.

TROPOMI Layer Height data are available on the NASA website (https://disc.gsfc.nasa.gov/ from 17[th] July 2023; last access: 12[th] April 2024. Older data were provided by the German Aerospace Center (DLR),

## 140 3.3 IASI

IASI is a Fourier transform spectrometer operating in the IR spectral domain. This instrument is located on both polar orbiting MetOp-B (IASI B) and MetOp-C (IASI C) satellites. The best spatial resolution at nadir is a circle which diameter is 12 km. Twice a day (measurements are possible both at daytime and nighttime), IASI are observing around 09:30 local time and around 21:30 local time (Clerbaux et al., 2009).

Sulfur dioxide observation data provided by IASI measurements are converted into a level 2 product by using the ULB-LATMOS retrieval algorithm (Clarisse et al., 2012). In our study, we use an optimal sulfur dioxide total vertical column computed from an estimated altitude of the volcanic plume. This estimation is based on another algorithm created for IASI sulfur dioxide plume altitude retrievals (Clarisse et al., 2014). IASI data are available at the AERIS data centre (https://iasi.aeris-data.fr/, last access: 19[th] September 2024).

## 150 3.4 OMI

OMI is a multispectral radiometer with spectral bands extending from the UV to the visible (VIS) domains. This sensor is carried by the polar orbiting Earth Observing System (EOS) Aura satellite. The best nadir spatial resolution is about 24 km x 13 km. Since 2011, this instrument has a two-day daily coverage with an overpassing at 13:45 local time (Qu et al., 2019).

OMI sulfur dioxide total vertical columns are used as independent observations to check the results of our assimilation ex-

periments. These total vertical columns are retrieved thanks to an algorithm based on a principal component analysis (PCA)





technique (Li et al., 2017). These vertical columns are computed by considering a hypothetical sulfur dioxide plume altitude located around 18 km. OMI data are available on the Earthdata Nasa website (https://omisips1.omisips.eosdis.nasa.gov/outgoing/OMSO2NRTb/, last access: 19th September 2024).

## 4 Description of the SO₂ assimilation experiments

MOCAGE (Modèle de Chimie Atmosphérique à Grande Echelle) is the CTM developed by the Centre National de Recherches Météorologiques (CNRM) at Météo-France (Josse et al., 2004). It has many operational uses such as air quality forecasting over France (Rouil et al., 2009) and over Europe with the contribution to the CAMS ensemble forecasting system (Marécal et al., 2015). MOCAGE is also used in an accident mode by the Volcanic Ash Advisory Centre of Toulouse (VAAC) when a volcanic eruption or an industrial accident occurs.

### 4.1 The model and its assimilation system


The CTM MOCAGE is a model using a semi Lagrangian scheme for the transport of chemical species which can be global or nested. It enables to predict chemical evolution of the atmosphere up to 4 days. In this study, we use MOCAGE on a 1 ° global domain with 47 hybrid $\sigma$ pressure levels distributed between the surface and 5 hPa (7 in the planetary boundary layer, 20 in the free troposphere and 20 in the stratosphere).

MOCAGE is an offline model and needs meteorological fields like wind speed and direction, temperature, humidity, pressure, rain, and clouds from a Numerical Weather Prediction (NWP) model or from a climate model. In this study, meteorological forcings are provided by the French NWP model Action de Recherche Petite Echelle Grande Echelle (ARPEGE) (Courtier, 1991; Bouyssel et al., 2022).

The model enables to transform species according to the chemical scheme RACMOBUS which is a combination of two chemical schemes. The first one, RACM, is computed for the tropospheric chemical reactions (Stockwell et al., 1997). It is completed with the sulfur cycle (Feinberg et al., 2019). The second chemical scheme, REPROBUS, is used for the stratospheric chemical reactions (Lefevre et al., 1994). Every 15 minutes, MOCAGE model provides the atmospheric composition of 112 gaseous species thanks to 379 chemical reactions and 57 photolysis reactions.

Both primary and secondary aerosols are modelled in MOCAGE (Guth et al., 2016), (Sič et al., 2015). Primary aerosols include desert dust, sea salt, black carbon, organic carbon and volcanic ash. In this study, volcanic ash modelling is turned off. Secondary inorganic aerosols are represented by sulfate, nitrate and ammonium aerosols. The aerosol size distribution is described by a sectional approach, with six size sections delimited by the following diameters: 0.002–0.01; 0.01–0.1; 0.1–1.1; 1.1–2.5; 2.5–10 and 10–50 $\mu$m . Desert dust and sea salt emissions depend on the wind strength and the type of the ground.


To forecast air quality, emissions of gaseous and aerosol species need to be taken into account by the model. Emission inventories are therefore used, such as the MACCity inventory for anthropogenic emissions (Lamarque et al., 2010) and the



MEGAN inventory for biogenic emissions (Sindelarova et al., 2014). Sulfur dioxide released into the atmosphere by passive degassing can be, as in our study, also part of the emissions included in the model (Lamotte et al., 2021). Daily emissions of

biomass burning provided by the Global Fire Assimilation System (GFAS) (Kaiser et al., 2012) are injected into the model at many vertical levels, depending on the latitude of fires (Cussac et al., 2020). Other species except lightning nitrogen oxides (Nox) (Price et al., 1997) and aircraft emissions (Lamarque et al., 2010) are emitted on the first five levels of the model (approximately 500m altitude).

Many species can be assimilated in MOCAGE. This is the case for gaseous species, such as $O_3$ (Emili et al., 2014), (El Aabaribaoune et al., 2021),(Vittorioso et al., 2024). Aerosols are also assimilated in MOCAGE (Descheemaecker et al., 2019), (Sič et al., 2016),(El Amraoui et al., 2022), (Cornut et al., 2023).

The assimilation system used in this study is the 3D-VAR, described hereafter. A short-range forecast from MOCAGE $x^b$

and observations $y$ are combined to find the optimal state $x^a$, taking into account their respective error covariance matrices B and R. $x^a$ can be searched as the sum of $x^b + \delta x^a$ where $\delta x^a$ is the increment minimising the cost function J:

$$J(\delta x) = \frac{1}{2}(\delta x)^T B^{-1}(\delta x) + \frac{1}{2}(y - \mathcal{H}[x^b] - \mathcal{H}[\delta x])^T R^{-1}(y - \mathcal{H}[x^b] - \mathcal{H}[\delta x]) \tag{1}$$

$\mathcal{H}$ is the observation operator used to obtain the model data in the observation space. Before running an assimilation experiment, a full description of R and B matrices is required. The background error covariance is spread on many vertical levels

and on many meshgrid thanks to the correlation matrix. In MOCAGE, the vertical and horizontal correlations are described as gaussian functions.

### 4.2 TROPOMI and IASI data assimilation setup

Several hourly 3D-VAR assimilations of volcanic $SO_2$ data have been conducted over the specific eruptive period from 9th April to 15th April 2021 into MOCAGE with a 1° horizontal resolution. Different simulations have been carried out, one

with the assimilation of IASI B and C (iasi_assim), one with the assimilation of TROPOMI (tropomi_assim), another with the assimilation of IASI and TROPOMI and the last one without assimilation (Table 1). The results of these experiments are compared to OMI observations.

In these experiments, IASI and OMI observations above 0.5 DU are used. This value corresponds to the lowest total columns measurable by these instruments (Koukouli et al., 2022), (Qu et al., 2019). For TROPOMI instrument, observation retrievals

with a $SO_2$ peak concentration at 15km of high are used. Moreover, an observation is used when the quality flag is above 0.5 and if the slant column is above 1 DU, matching the noise of the instrument.

Averaging kernels are taken into account in this study. For TROPOMI, averaging kernels are only given for the a priori profiles from the TM5 CTM. Nevertheless, averaging kernels for the other a priori profiles can be estimated by multiplying them by a scaling factor (Theys, 2018) as described by the following equation:





| Experiment | Assimilation of TROPOMI | Assimilation of IASI B and C | Assimilation of OMI |
|---|---|---|---|
| iasi_assim | No | Yes | No |
| tropomi_assim | Yes | No | No |
| joint_assim | Yes | Yes | No |
| dry | No | No | No |

**Table 1.** Description of the performed experiments during this study with the different assimilated instruments.

$$\text{AVK(z)} = \frac{\text{AVK}_{TM5}(z) \times \text{AMF}_{\text{TM5}}(z)}{\text{AMF}_{\text{15km}}} \tag{2}$$

with AVK(z) the averaging kernels at a given altitude, $\text{AVK}_{TM5}(z)$ and $\text{AMF}_{\text{TM5}}(z)$ the averaging kernels and the air mass factor at the altitude z from the TM5 CTM and $\text{AMF}_{\text{15km}}$ the air mass factor for the a priori profile containing a peak at 15km of altitude.

For IASI, averaging kernels are not given in the observation files. However, they can be computed at many heights thanks to
the SO$_2$ total columns, according to the following equation:

$$\text{Avk(z)} = \frac{\text{Y(z)}}{\text{Y(z}_{\text{ref}})} \tag{3}$$

with z the hypothetical SO$_2$ injection altitude, Avk(z) the averaging kernels at a given altitude z, Y(z) the total columns computed for a SO$_2$, injection at the altitude z and Y(z$_{\text{ref}}$) the total column computed for a reference altitude injection. In our case, this altitude is provided by the observation files.

In this study, we assume there is no spatial correlation in the observation error. For TROPOMI, observation error covariance is directly computed from satellite data. The uncertainties of IASI measurements vary according to the value of the total column measured. In the case of this eruption, IASI measured total SO$_2$ columns ranging from 0.5 to 20 DU. The uncertainties in this range of observations vary from around 25% to 5% of the observation (Clarisse et al., 2012). For IASI, we set the observation error standard deviation to 15% of the observation values. Correlation matrix is the same in all experiments with
data assimilation.

For observations stronger than 20 DU, TROPOMI Layer Height product is able to diagnosed the altitude of the plume. During this volcanic eruption, 90% of the diagnosed heights are between 9 and 21 km. Consequently, to force SO$_2$ injection between these two altitudes, we chose a profile containing strong values (1e-8 ppv) between altitudes of 9 and 21 km as back-
ground error standard deviation.

As in the operational mode of MOCAGE, a forecast initialised by the assimilation outputs is launched at 00 UTC each day. In our study, this forecast is performed up to a 72 h term range.



## 5 Impact on analyses

### 5.1 Impact on SO₂ and sulfate

The assimilation of SO$_2$ total column measurements significantly enhances MOCAGE's ability to describe SO$_2$ plumes. When no observations are assimilated, no SO$_2$ plume is represented by the model. However, when satellite observations are assimilated, a SO$_2$ plume is simulated and corrected more or less frequently depending on the instrument or combination of instruments and thus on the overpass time of the corresponding satellites.

Figure 1 shows the SO$_2$ total columns observed and analysed on 10$^{th}$ April 2021 at various times (02, 14, 16 and 17 UTC). The observations are depicted on the first line for TROPOMI, the second and the third lines for IASI B and C respectively. The model's outputs are shown on the fourth line for the iasi_assim experiment, on the fifth line for the tropomi_assim experiment and on the sixth line for the joint_assim experiment. At 02 UTC, the tropomi_assim simulation does not present a SO$_2$ plume, in contrast to the plume modelled with IASI assimilation. Nevertheless, the model underestimates the total column values against the IASI observations. Due to the use of UV wavelengths, TROPOMI is unable to measure SO$_2$ total column during night. Consequently, no SO$_2$ plume is modeled in MOCAGE until 17 UTC. Before the overpass of TROPOMI, IASI instruments measure SO$_2$ total columns once again. It allows to increase both intensity and size of the plume in MOCAGE at 14 UTC. At 17 UTC, a plume appears in the tropomi_assim experiment thanks to the TROPOMI overpass. Compared to the TROPOMI observations, high total columns values are underestimated by MOCAGE. At this time, the simulated volcanic plume in the iasi_assim experiment is slightly smaller and weaker than the plume in the tropomi_assim experiment and the TROPOMI observations. Assimilation of TROPOMI data allows to simulate a strong area value in vicinity of the volcano in both tropomi_assim and joint_assim experiments. This structure of strong values is not modeled with the IASI data assimilation because it corresponds to the latest volcanic SO$_2$ emission. This new release, due to a new eruption event, took place between the IASI and the TROPOMI overpasses. Simulated SO$_2$ total columns in joint_assim experiment are stronger, in particular around $55°W$ where observations are strong. For this part of the plume, the difference between the model and the observations is weaker because it is analysed five times during this day whereas the model is corrected four times when IASI is assimilated and only once when TROPOMI is assimilated. The greater the number of model corrections, the smaller the differences between observations and the model. In this experiment, the shape of the plume is slightly different with a pattern around 39°W which is simulated with the IASI data assimilation and not with the TROPOMI one.

Figure 2 shows the SO$_2$ total columns observed and analysed on 11$^{th}$ April 2021 at 11, 13, 17 and 18 UTC. Until 17 UTC not included, the SO$_2$ plumes simulated by the iasi_assim and the joint_assim experiments are similar. At 11 UTC, the eastern parts of the plume are consistent between experiments. Thanks to the IASI overpass at the end of the previous day, the plume is closer to the volcano and SO$_2$ total columns of the western part of the plume are stronger compared to the tropomi_assim experiment. At 13 UTC, IASI SO$_2$ total columns are assimilated, reducing the SO$_2$ plume over the ocean and increasing the values in the vicinity of the volcano into the model. At 17 UTC, TROPOMI observations are assimilated and the shape of the SO$_2$ plume is consistent between the experiments. The SO$_2$ plume is more extended with the assimilation of TROPOMI.



Concerning the values of the total columns, strong values areas are more intense with the assimilation of both instruments, particularly near the volcano and in the middle of the Atlantic Ocean. At 18 UTC, the OMI overpass enables to validate these
experiments with independent observations. The $SO_2$ plume simulated by the joint_assim experiments seems to be the closest to OMI observations. Indeed, in this experiment, the strong $SO_2$ total columns patterns match better to the observations, in particular around $39°W$ and around $55°W$.

The vertical cross-section at 13.5°N, illustrated in figure 3, presents the vertical $SO_2$ concentrations at various times on 11th April 2021 (11, 13, 17 and 18 UTC). The chosen assimilation settings result in a plume that extends from 9 to 21 km in
altitude. Until 17 UTC not included, the altitude of the maximum $SO_2$ concentration in the different experiments is consistent with an altitude around 13 km to 17 km. The lowest concentrations are simulated in the tropomi_assim assimilation. At 11 UTC, the IASI assimilation of the end of the previous day enables to simulate a plume located closer to the volcano compared to the tropomi_assim experiment. At 13 UTC, with the assimilation of IASI, a new strong concentration pattern appears near the volcano around 13 km of altitude. At 17 UTC, in experiments where TROPOMI observations are assimilated, $SO_2$
concentration increases around 9km of altitude in vicinity of the volcano The altitude of the plume is closer to the TROPOMI Layer height product in the joint_assim experiment. However, around $50°W$, TROPOMI Layer Height product shows a plume between 10 km and 16 km of altitude. In the experiments, the height of the plume varies between 15 km and 20 km, which is too high compared to the TROPOMI Layer Height Product. Nevertheless, around $50°W$, few observations meet the criteria for TROPOMI_LH to be able to diagnose the height of the plume. It is difficult to conclude whether the plume altitude is correctly
represented in the model when there are few or no observation above 20 DU.

$SO_2$ can be converted into sulfate aerosol in the presence of water vapour. This process is modelled in MOCAGE. Indeed, sulfate total columns show structures in the assimilation experiment which are not shown without assimilation.

Figure 4 shows the $SO_2$ total columns on top panel, the sulfate total columns on the middle panel and the Aerosol Optical Depth (AOD) on the bottom panel on 14th April 2021 at 07 UTC for different experiments with assimilation (iasi_assim,
tropomi_assim and joint_assim) and without assimilation (dry). This figure shows strong differences between each experiment. Indeed, at this date, the analysed $SO_2$ total columns are stronger by assimilating TROPOMI instrument than in the iasi_assim and the joint_assim experiments. Nevertheless, $SO_2$ total column values never reach 3 DU. Without assimilation, no $SO_2$ plume is modelled by MOCAGE.

The assimilation of volcanic $SO_2$ total columns allows the model to simulate sulfate aerosols (figure 4 on the middle panel).
Indeed, a sulfate plume is simulated from the volcano to Guinea. This sulfate plume is not modelled in the experiment without assimilation. In the tropomi_assim experiment, strong sulfate total columns are simulated in French Guiana and in Venezuela. In this area, values exceed 25 mg/m$^2$ whereas values do not reach 20 mg/m$^2$ in iasi_assim and joint_assim experiments. Elsewhere the modelled sulfate total columns are consistent with from one assimilation experiment to another.

Total AOD are slightly increased in Central America with the $SO_2$ assimilation (figure 4 on fifth line). This rise is more important by assimilating TROPOMI in French Guiana and in Venezuela where AOD reach 0.5 against 0.4 in the iasi_assim and in the joint_assim experiments and 0.3 without assimilation (figure 4 on fourth line). However, few AOD observations





from MODIS (Moderate-Resolution Imaging Spectroradiometer) and VIIRS (Visible Infrared Imaging Radiometer Suite) instruments are available in this area. This makes it impossible to validate AOD assimilation.

### 5.2 Impact of the assimilation on the detection of $SO_2$ threshold exceedances

To assess the impact of assimilation on volcanic $SO_2$, the number of grid cells reaching 1 DU and 5 DU thresholds, over a sub-domain extending from 90°W to 40°E and from 20°S to 30°N, has been calculated and plotted on the figure 5 for the simulations: iasi_assim in red, tropomi_assim in blue and joint_assim in green, and for the observations: TROPOMI in orange, IASI B in purple, IASI C in magenta and OMI in grey. Since there are often several observations per grid cell, we looked at the number of grid cells where the minimum and maximum of the total columns exceed these thresholds. These values are represented by horizontal lines. The number of grid cells where the median of the observations is superior to these thresholds is shown by a dot. Finally, the limits of the bars represent the number of grid cells where the 25th and 75th quantiles exceed the thresholds.

Between the 9th April and the 11th April 2021, the number of meshes where the median of observations exceeded 1 DU is consistent between instruments. After this date, there were more grid cells where the median number of observations exceeded 1 DU with TROPOMI. Moreover, with this instrument, the dispersion of observations is high. There are few grid cells where all the observations exceed 1 DU, whereas on 14th April, there were more than 1200 grid cells where the maximum number of observations reach 1 DU. This dispersion is lower for instruments with lower resolutions. For IASI, the number of grid cells where the median of observations is superior or equal to 1 DU varies less than with UV instruments. This can be explained by IASI's high sensitivity to water vapour, which masks part of the $SO_2$ column.

In the assimilation experiments, the number of points where the total column reach 1 DU is identical between iasi_assim and joint_assim until the end of the day on 10 th April 2021. Until this date, the number of grid cells where the total column reaches 1 DU is zero with the TROPOMI assimilation. After this date, the number of grid cells where the total column exceeds 1 DU is always bigger in the joint_assim experiment. The number of grid cells above 1 DU is lower when only TROPOMI is assimilated in the morning until 11th April. This number is smaller when only IASI is assimilated. The differences in the number of grid cells exceeding 1 DU are greater at the end of the study period. Until 11th April 2021 in the experiments where TROPOMI was assimilated and until 12th with the assimilation of IASI alone, the number of grid cells where the model exceeded 1 DU was consistent with the number of grid cells where the median of the observations exceeded this threshold. After this date, the number of grid cells exceeding 1 DU exceeds the number of meshes where the 75th quantile of TROPOMI observations exceeds 1 DU. Whatever the instruments used, the number of points above 1 DU is too high compared to IASI and OMI observations. This shows that the extension of the $SO_2$ plume is too large in the model.

The number of grid cells with observations exceeding 5 DU is lower and similar for each instrument. On 9th, on 14th and on 15th April 2021, none of the observations exceeded this threshold. In the model, no column exceeds 5 DU for these dates. Generally, the number of $SO_2$ columns exceeding 5 DU in the model is similar to the number of grid cells where the median of observations exceeds 5 DU. Between the end of the day on 10th and 12th April, the number of meshes is slightly greater by



assimilating the 2 instruments and is closer to the number of meshes where the median of the OMI observations reaches 5 DU even if the number of points above 5 DU is slightly underestimated at the end of the day on 10th April 2021. On this day, this number is underestimated in the model because a new eruption took place between the last assimilation of TROPOMI and the

overpass of OMI. On 11th April, the TROPOMI assimilation added a significant number of points above 5 DU in the model because of a large number of TROPOMI observations above 5DU. On 12th April, the extension of the plume reaching 5 DU is greater when assimilated in the tropomi_assim experiment. In fact, the value of the total columns fells in the model thanks to the assimilation of IASI. The TROPOMI overpass at the end of the afternoon also reduces the modelled total columns. The number of points above 5 DU in the model becomes similar.


To assess the accuracy of the model in simulating $SO_2$ total columns, a threshold-based analysis was implemented. The goal was to determine the number of instances where both the observations and the model successfully identified $SO_2$ total columns above certain thresholds (labelled as Hits), as well as the instances where the observations exceeded these thresholds but the model failed to detect them (labelled as Misses). Using these metrics, we defined the Probability of Detection (POD),

a ratio that ranges from 0 to 1. The POD is calculated by dividing the number of Hits by the sum of Hits and Misses for a given threshold. A POD score of 1 indicates a perfect detection by the model, meaning that all observed instances above the threshold were correctly simulated. On the other hand, a POD of 0 signifies that none of the observed $SO_2$ total columns above the threshold were detected by the model. The POD is computed with the following equation:

$$POD = \frac{Hits}{Hits + Misses} \qquad (4)$$

Figure 6 shows the Probability of Detection (POD) computed for 1 DU and 5 DU thresholds against TROPOMI, IASI and OMI observations. Dots represent times when there is no observation. Crosses represent the moments when simulated $SO_2$ total columns are under a threshold whereas some observations exceeds this threshold. Against TROPOMI instrument, POD values are generally better in the experiments in which TROPOMI observations have been assimilated. In these experiments, POD values exceed 0.75 until the 13th April. The POD values are over 0.75 until 11th April in iasi_assim experiment. POD

values decrease at the end of the study period. For the 5DU threshold, POD values are slightly higher in joint_assim experiment, especially on 10th and 11th April, when around 100 TROPOMI observations exceed 5 DU. On 12th April, POD values are around 0.25 in the experiments in which TROPOMI instrument is assimilated. The model did not $SO_2$ total columns higher than 5 DU for this date in iasi_assim experiment. Between the 13th and the 15th April and on 9th April, no $SO_2$ total column above 5 DU is simulated in MOCAGE. For these days, between 1 and 9 observations above 5DU are measured by TROPOMI.


POD values, computed for a 1 DU threshold and with IASI observations, exceed 0.9 until 12th April in experiments in which IASI instruments are assimilated. No $SO_2$ total column is simulated with the TROPOMI assimilation until 10th April because TROPOMI overpasses the plume after IASI. In the morning of 9th April, no observation above 1 DU is detected by IASI. From 11th April, POD values are not null in the tropomi_assim experiment but they are smaller than the one calculated





from the iasi_assim and joint_assim experiments. In the afternoon of $9^{th}$ April, only one observation above 5 DU is measured by IASI. At this location, the total column is under 5 DU. For these threshold and compared to tropomi_assim experiment, the probability to simulate high $SO_2$ total columns increases thanks to the IASI assimilation. Despite numerous observations above 5 DU, many events are missed on $10^{th}$ April with a POD reaching nearly 0.4 in the morning and 0.5 in the afternoon. Most of simulated $SO_2$ total columns are between 1 DU and 5 DU. The maximum of POD is obtained on $11^{th}$ April after the

assimilation of many observations measured by IASI exceeding 5 DU on $10^{th}$ and on $11^{th}$ April.

    When comparing with OMI observations, the POD values computed with a 1 DU threshold are often consistent between the tropomi_assim and the joint_assim experiments. POD values are slightly better in joint_experiment on $11^{th}$ and $12^{th}$ April. One exception occurs on $9^{th}$ April when POD value is higher by assimilating IASI instruments. For the 5 DU threshold, POD value is greater with the joint_assim experiment on $10^{th}$ and $11^{th}$ April. On $12^{th}$ and $13^{th}$ April, no $SO_2$ total column above 5

DU is modelled by MOCAGE whereas OMI measured observations above this threshold. Elsewhere in the study period, no observation greater than 5 DU is measured by OMI instrument and modelled by MOCAGE.

    In the various modelling experiments, assimilating $SO_2$ total column data enhances the performances of the CTM MOCAGE by enabling the simulation of a $SO_2$ plume. However, for the lowest concentration threshold, the simulated $SO_2$ plume tends to be overly extensive, and the corresponding $SO_2$ burden is too high when compared to observational data. Nevertheless, for

higher thresholds, $SO_2$ plume area and $SO_2$ burden are consistent with the observations in the experiments where TROPOMI instrument is assimilated. Especially for the strong thresholds, POD values show an improvement of the model when both IASI and TROPOMI instruments are assimilated.

## 6   Impact of assimilation on forecasts

    In this part, we study the impact of the assimilation on forecasts. To initialise the forecast, we use the assimilation outputs from

the joint_assim experiment. We use the term D0 for a 24h range term forecast, D1 for a 48h range term forecast and D2 for a 72h range term forecast.

    On the figure 7, the fourth and the fifth line show $SO_2$ total column forecast for the $12^{th}$ April 2021 at 01, 13, 15, 16 and 17 UTC. These lines represent respectively a forecast initialised by the $11^{th}$ April 2021 analysis outputs and by the $10^{th}$ April 2021 analysis outputs. $SO_2$ total column observations are plotted on the first line for TROPOMI, the second and the third line

for IASI and on the last line for OMI. When the model forecasts are initialised by the $9^{th}$ April analysis outputs, no $SO_2$ total column value greater than 1 DU is present because even if the eruption started on $9^{th}$ April, none of the assimilated instruments overpass the plume on this day. Using the results of the analysis on $10^{th}$ April enables to the model to simulate a $SO_2$ plume which reaches West Africa. The observations show a plume reaching Africa but also the vicinity of the volcano. The western part of the plume is not modelled by MOCAGE on $12^{th}$ April 2021 with the use of $10^{th}$ April analysis outputs because the last

assimilation took place almost 2 days before. The more recent $SO_2$ emissions can not be simulated. Nevertheless, the part of the already simulated plume matches well with the observed plume intensity. Using the latest available analysis outputs from $11^{th}$ April, MOCAGE predicts a plume which shape is closer to the observed one but it is always smaller. However, the model



tends to overestimate the total intensity of the $SO_2$ column compared to the TROPOMI and IASI measurements. In addition, using the latest available analysis predicts a $SO_2$ plume closer to the volcano.

Figure 8 represents the number of grid cells exceeding 1 DU and 5 DU for the D0 forecast in blue, for the D1 forecast in red and for the D2 forecast in green. Number of grid cells exceeding these thresholds are computed for TROPOMI in orange, IASI B in purple, IASI C in magenta and OMI in grey. We looked at the number of grid cells where the minimum and maximum of the total columns exceed these thresholds. These values are represented by horizontal lines in the figure 8. The number of grid cells where the median of the observations exceeds these thresholds is shown as a dot. Finally, the limits of the bars represent

the number of grid cells where the 25 and 75 quantiles exceed the thresholds.

Generally speaking, the number of meshes exceeding 1 DU or 5 DU decreases with the forecast period. This was not observed on 9[th] and 10[th] April, when no mesh exceeded 1 DU. The D0, D1 and D2 forecasts show the presence of a plume from 11[th], 12[th] and 13[th] April 2021. These forecasts are initialised by the output of the assimilation of 9[th] April, i.e. before the beginning of the eruption. On 11[th] April, there were around 200 grid cells where the model exceeded 1 DU for the D0 forecast. This number corresponds to the minimum number of grid cells where the TROPOMI and OMI observations reach 1 DU and is below the number of grid cells where the median of the IASI observations reaches 1 DU. On 12[th] April, the plume forecast with D0 was larger, with around 500 meshes exceeding 1 DU, corresponding to the number of meshes where the 25[th] quantile of the TROPOMI observations reached 1 DU and also where the median of the OMI observations reached

this threshold. After this date, the number of occurrences of the total column in $SO_2$ exceeding 1 DU increases and becomes greater than the number of grid cells where the IASI and OMI observations reach 1 DU. However, this number remains smaller than the maximum number of meshes calculated using TROPOMI observations. The number of points where the total column reaches 1 DU decreases with the forecast term range. Nevertheless, this number always exceeds the number of grid cells where the OMI and IASI observations reach 1 DU from 14 April onwards. Regarding the 5 DU threshold, no grid cell exceeds this

threshold for the D2 forecast. The D1 forecast shows a low number on 12[th] April when the total columns reach 5 DU. However, this number is similar to the number of grid cells where the median of TROPOMI and IASI observations reaches 5 DU. In addition, for this day, the number of points where the model reaches 5 DU is similar between the D0 and D1 forecasts. For 11[th] April, the number of occurrences of a total column greater than or equal to 5 DU is low in the model compared with the UV instruments. This number is within the range of grid cells where IASI observations exceed 5 DU.

The figure 9 represents the POD metric calculated by comparing the observations and the forecasts at several time steps for 1 DU and 5 DU thresholds. Whatever the thresholds used, in the D0 forecast, plume is well represented with the strongest POD values even if the POD values decrease compared to the obtained values computed with the analysis outputs. It indicates a rise of the number of Misses events. For the lowest threshold and for a D0 forecast, POD values reach at maximum 0.6 on 12[th] April 2021 and 13[th] April 2021 compared to TROPOMI, nearly 0.75 on 12[th] April 2021 compared to IASI and 0.7 on 13[th]

April compared to OMI. These values become lower with the increase of the forecast range except on 15[th] April and 16[th] April with TROPOMI. For larger threshold, Misses events dominate the 24 h and the 48 h forecasts. Moreover, POD values decrease in the D0 forecast when the threshold value increases. The forecasts initialised by the assimilation outputs are improved, in




particular when the most recent analysis is used to forecast and for lower $SO_2$ total columns. Nevertheless, for higher $SO_2$ total columns, MOCAGE encounters difficulties in accurately modelling such intense $SO_2$ total columns.

Incorporating $SO_2$ total column analysis outputs in the MOCAGE forecasts enhances its ability to model $SO_2$ plume. In the absence of the assimilation, the model fails to forecast the presence of a $SO_2$ plume. The use of recent analysis outputs progressively aligns the predicted location of the plume with the observations. However, the predicted intensity of the plume remains subject to a high uncertainty.

## 7   Conclusion and perspectives

In this paper, we study the input of the assimilation of volcanic $SO_2$ total columns from TROPOMI, IASI and both TROPOMI and IASI into the CTM MOCAGE in the case of the La Soufrière Saint-Vincent eruption between $9^{th}$ and $15^{th}$ April 2021. For the background error covariance matrix, we used a profile containing high values in the volcanic plume vertical range extension. We considered a plume ranging from 9km to 21km, corresponding to 90% of the $SO_2$ plume heights diagnosed by the TROPOMI Layer Height product.


    Thanks to the assimilation of TROPOMI and IASI instruments, a $SO_2$ plume is simulated in MOCAGE. This plume is modelled more or less early and corrected more or less often, depending on the time of the satellite overpass and on the instrument technology. The assimilation of both TROPOMI and IASI instruments leads to a larger plume and a more important amount of $SO_2$ in the model. During this eruption, MOCAGE is able to simulate the process of converting $SO_2$ into sulfate aerosols.

A sulfate plume, stronger by assimilating only TROPOMI, is computed by MOCAGE. With this creation of sulfate aerosols, AOD increases slightly. Nevertheless, few AOD observations are available during the studied period. Compared with $SO_2$ total columns observations, the number of pixel stronger than 1 DU in the model is too large but the probability to detect a $SO_2$ total column is important. In the model, number of points with a total column stronger than 5 DU is close to the number of grid cells with a median observation stronger than 5 DU, especially when IASI and TROPOMI are assimilated. Compared to

independent observations from OMI, the probability to detect values stronger than 5 DU is better is better when assimilating observations from both instruments.

    Using assimilation outputs to compute forecasts improves the representation of $SO_2$ total columns in the model. The size and the shape of the plume depends on the forecast range term. The more the forecast range term is small, the more the plume

size is important. Sometimes and especially for low values of $SO_2$ total column, the size of the modelling plume is too large compared to the observations. Concerning the probability to detect a $SO_2$ total column stronger than a threshold, it is generally better for the short term forecast.

    One potential source of discrepancy is the assumptions regarding the altitude and thickness of the plume, which are dynamic

in space and time. An inaccurate representation of the $SO_2$ plume height within the model could lead to a more dispersed plume



or, depending on wind shear conditions, could result in the plume drifting in an incorrect direction. To refine our assimilation process, we could incorporate observed volcanic SO$_2$ plume heights from both IASI and TROPOMI. This approach, as suggested by (Inness et al., 2021), would likely yield a more accurate simulation of the altitude of the plume, thereby producing a modelled plume shape that better reflects reality.


Another factor contributing to the overestimation of the plume extent within the MOCAGE model is the assimilation settings. Specifically, the application of an excessively large horizontal and vertical correlation length could lead to a too extended plume both on the vertical and the horizontal dimensions. Moreover, the chosen standard deviation for the background error can significantly influence the simulated intensity of the plume. With a too large value, the $SO_2$ plume concentration can be
underestimated in MOCAGE whereas with a too small value, the plume intensity can be too strong.

Additionally, inherent uncertainties in the model itself must be considered. The chemical processes involving SO$_2$, including reactions that may not be fully captured, introduce additional complexity into the simulation. Furthermore, the meteorological data driving the model are not without their uncertainties, which can compound the challenges in accurately modelling the transport and transformation of SO$_2$ emissions.


Satellites can give information on the locations where no SO$_2$ is detected. The use of this information for the assimilation could improve the process because it allows to limit the shape of the plume.

To facilitate the assimilation of the SO$_2$, particularly the convergence of the minimizer, we could also use a prior volcanic
SO$_2$ emission. In fact, the model and the observations are very different, sometimes by several orders of magnitude. To estimate volcanic SO$_2$ emissions, a source inversion of the volcanic SO$_2$ could be used (Boichu et al., 2013).

We also have seen that the more frequently the model is corrected, the closer it is to the observations. Assimilating additional instruments would therefore improve assimilation of the SO$_2$. The best option would be to assimilate observations from
geostationary satellites covering the globe.

Finally, we made a specific adjustment for this eruption. Ideally, we would like to have a task running daily with assimilation settings that would allow us to assimilate the volcanic SO$_2$ for each eruption.

*Code availability.* The code used to generate the analysis (MOCAGE and its variational assimilation suite) is a research-operational code property of Météo France and CERFACS and is not publicly available yet. The readers interested in obtaining parts of the code for research
purposes can contact the authors of this study directly.



*Data availability.* All results are available upon request to the authors. Relevant model outputs used to draw the figures will be uploaded to Zenodo shortly.

*Author contributions.* MB and VG designed this study. MB carried out and interpreted the simulations with JA during his internship supervised by MB and VG. MB is the main contributor to the manuscript. VG and JA reviewed and contributed to the manuscript.

*Competing interests.* No competing interest.

*Acknowledgements.* We want to thank Emanuele Emili for the scientific advice and Pascal Hedelt for providing TROPOMI Layer Height product and give information in the use of this product.



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





**Figure 1.** Observations assimilated and analyses of SO₂ total columns on 10th April 2021 at 2, 14, 16 and 17 UTC. The first three rows correspond respectively to TROPOMI, IASI B and IASI C observations. Analysis outputs are plotted on the fourth line for iasi_assim experiment, on the fifth line for tropomi_assim experiment and on the sixth line for the joint_assim experiment. The shaded areas shown on the first 3 lines correspond to areas where there are no observations or where observations have not been assimilated. Observations are not assimilated when they are less than 0.5 DU for IASI and TROPOMI's slant columns are less than 1 DU.





**Figure 2.** Observations and analyses of SO$_2$ total columns on 11$^{th}$ April 2021 at 11, 13, 17 and 18 UTC. The first three rows correspond respectively to TROPOMI, IASI B and IASI C observations. The last row corresponds to OMI observations. Analysis outputs are plotted on the fourth line for iasi_assim experiment, on the fifth line for tropomi_assim experiment and on the sixth line for the joint_assim experiment.



**Figure 3.** Vertical sections of analysed SO$_2$ concentration at 13.5°N latitude on 11$^{th}$ April 2021 at 11, 13, 17 and 18 UTC. Rows correspond respectively to the TROPOMI data assimilation, IASI assimilation, joint assimilation and the height of SO$_2$ plume provided by the TROPOMI Layer Height product.

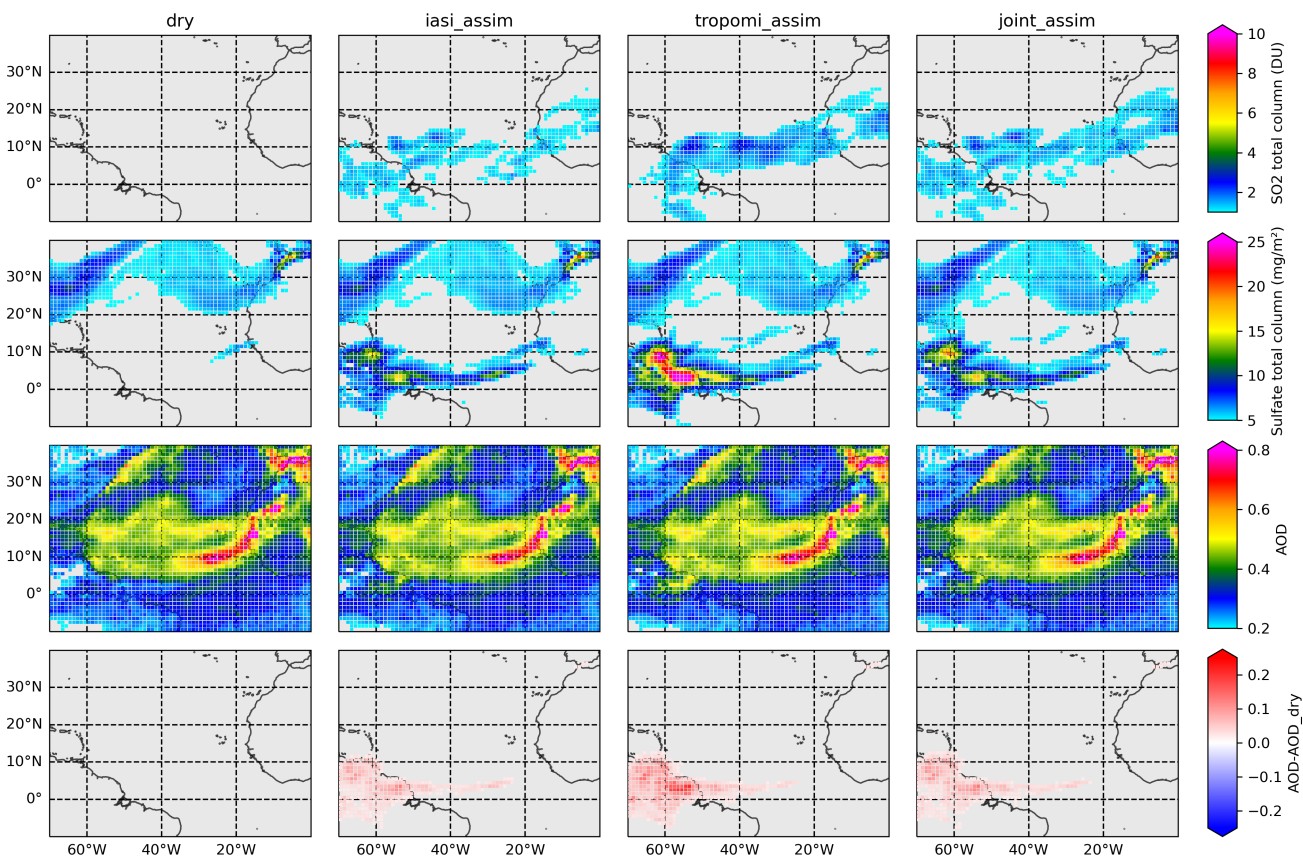

**Figure 4.** SO$_2$ total column, sulfate total column, AOD and difference between AOD and AOD of dry experiment on 14[th] April 2021 at 7 UTC for dry, iasi_assim, tropomi_assim and joint_assim experiments.



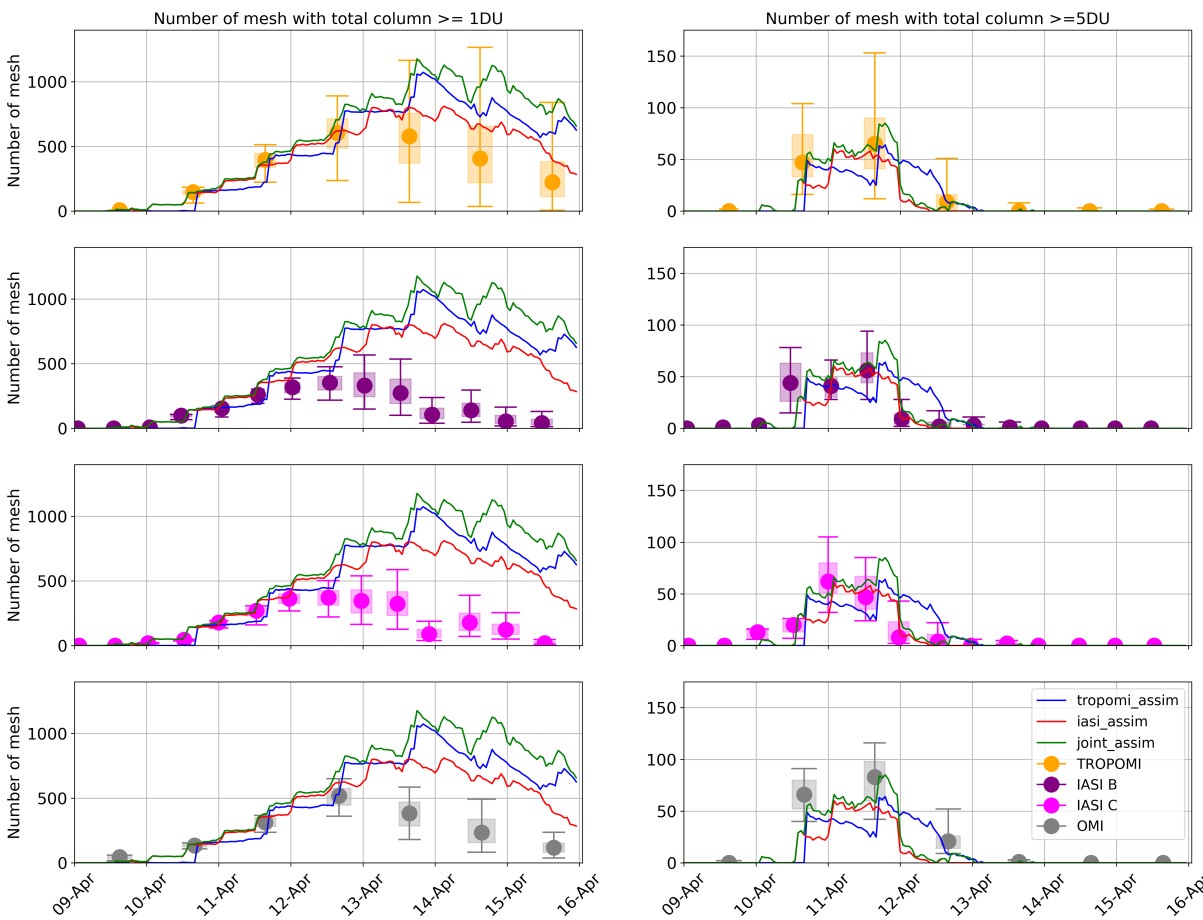

**Figure 5.** Number of grid cells where analyses exceed 1DU and 5DU. The blue, red and green lines show the number of points at which the total columns reach these thresholds in the tropomi_assim, the iasi_assim and the joint_assim experiments. Orange, purple, magenta and grey boxplots represent the number of grid cells where TROPOMI, IASI B, IASI C and OMI observations exceed the threshold.





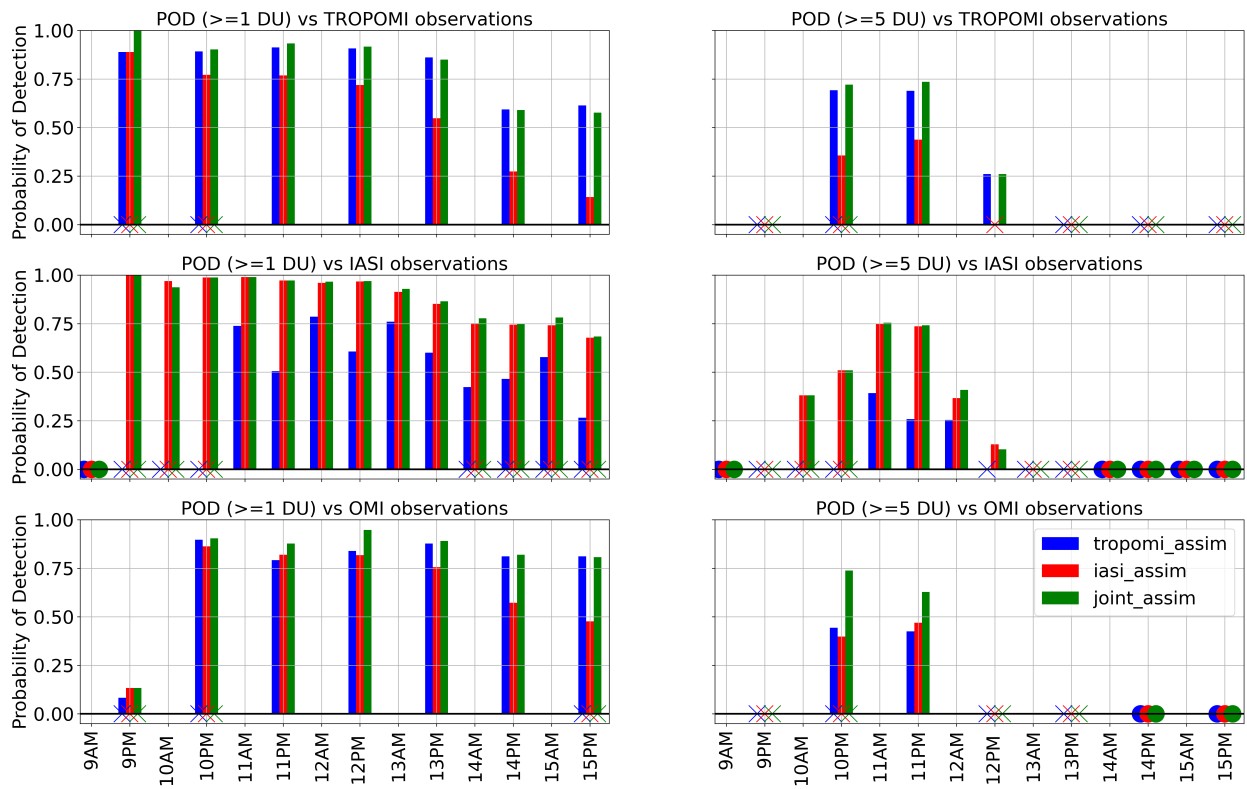

**Figure 6.** Probability of detection for 1 and 5 DU thresholds for the three experiments: tropomi_assim in blue, iasi_assim in red and joint_assim in green. Dots represent times when there is no observation. Crosses represent the moments when there are only misses events.





**Figure 7.** Observations and forecasts of SO$_2$ total columns on 12$^{th}$ April 2021 at 01, 13, 15, 16, 17 UTC. The first row and the last row correspond to TROPOMI and OMI observations. Forecasts are computed from the 11$^{th}$ April analysis outputs (2$^{nd}$ row) and from the 10$^{th}$ April analysis outputs (3$^{rd}$ row). These forecasts are available on 12$^{th}$ April





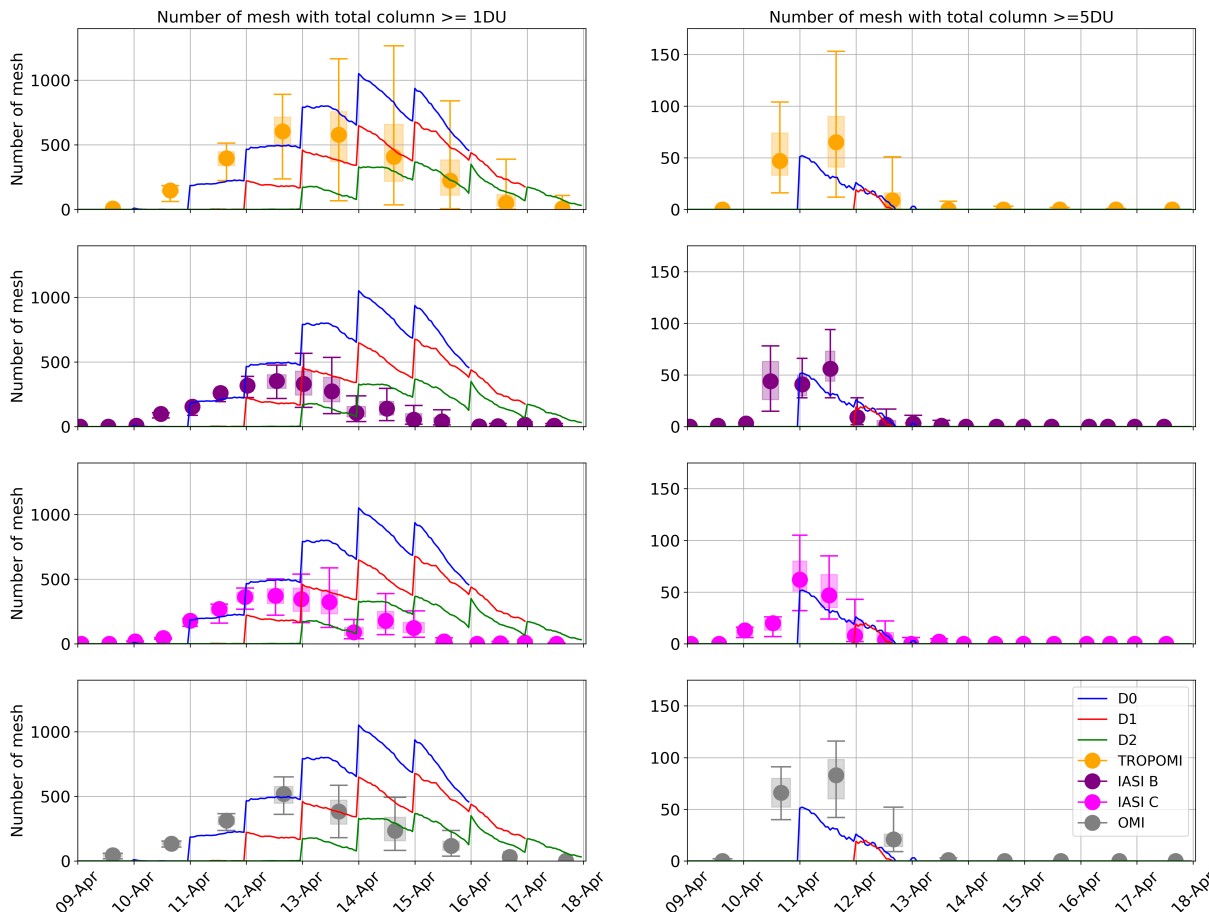

**Figure 8.** Number of grid cells where forecasts initialised by the joint_assim outputs reaches 1 DU and 5 DU. The blue, red and green lines show the number of points at which the total columns reach these thresholds in the D0, D1 and D2 forecasts. Orange, purple, magenta and grey boxplots represent the number of grid cells where TROPOMI, IASI B, IASI C and OMI observations exceed the threshold.





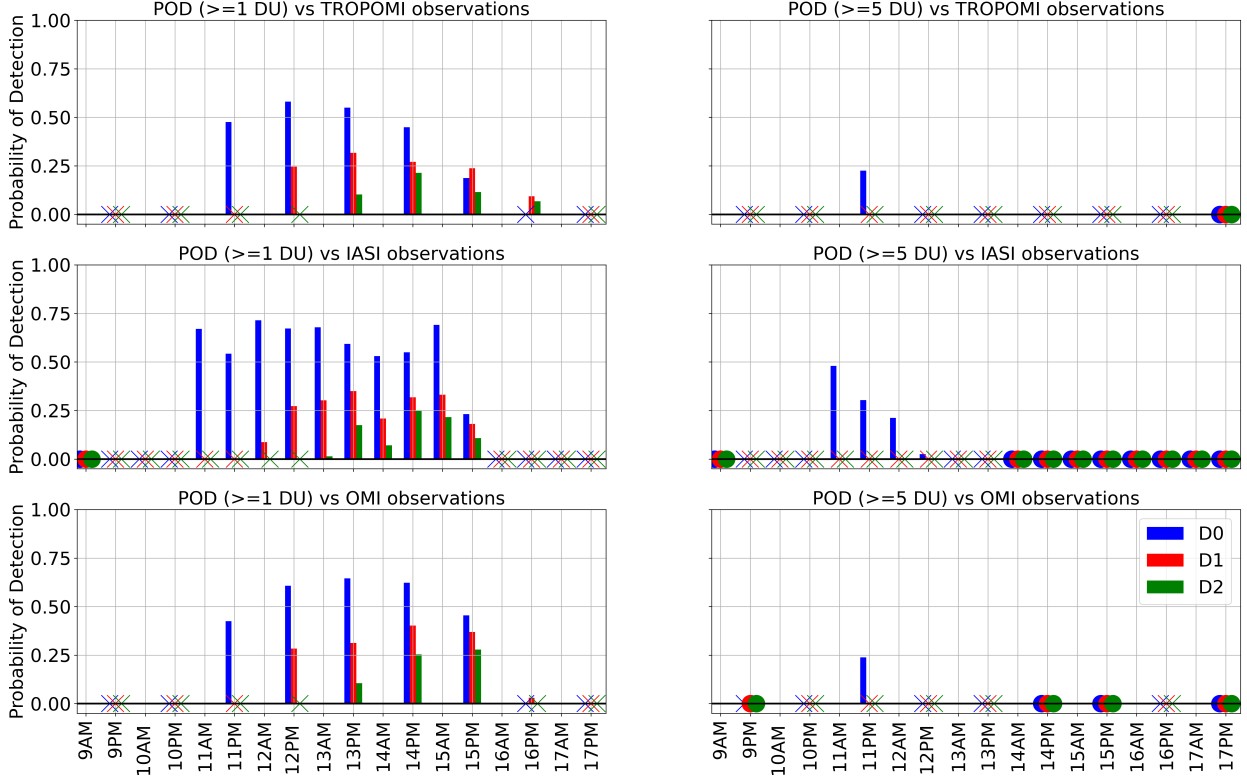

**Figure 9.** Probability of detection for 1 DU and 5 DU thresholds for the D0 forecast in blue, the D1 forecast in red and the D2 forecast in green. Dots represent times when there is no observation. Crosses represent the moments when there are only misses events.