# Peer review of "Assimilation of volcanic sulfur dioxide products from IASI and TROPOMI into the chemical transport model MOCAGE: case study of the 2021 La Soufrière Saint-Vincent eruption"

_EGUsphere, 2024_

## Referee Comment (RC2)

Review of GMD-2024-2941 « Assimilation of volcanic sulfur dioxide products from IASI and TROPOMI into the chemical transport model MOCAGE: case study of the 2021 La Soufrière Saint-Vincent eruption "

In this article, the authors assimilate observations of $SO_2$ total column from two different sensors (TROPOMI and IASI) into the MOCAGE CTM, in order to provide forecasts of volcanic plume for the VAACs. The topic is of importance, and the paper is well written and easy to read. The authors show a good command of the subject, and the article show promise, but I think it can be improved further. I have a few important questions, which amounts to a major revision. These key points are, in my opinion:

- The system described here has been designed to work only for this eruption, as mentioned by the authors themselves in the conclusion, because of the setting of the background error standard deviation (lines 239-240). If applied, say, to the Pinatubo eruption, adding increments between 9 and 21 km altitude will probably give a very wrong profile. Isn't it possible to design a system that doesn't need the manual input of this key information about injection height? Ideally, aerosol layer height information should be used.

- The main objective of the MOCAGE CTM in the VAACs is to provide forecasts of the aerosol layer (ash or sulphuric acid). I understand that ash in not in the scope of this article. However, the aerosol validation aspect Is treated very shortly, as compared to the $SO_2$ aspect. The volcanic signal is small as compared to the tropospheric aerosol signal for this eruption, which complicates things. The authors should try to either find and compare against retrievals which clearly show a volcanic signal for this eruption (OMPS-LP possibly), or chose another eruption for which the aerosol signal has been extensively documented, such as the Hunga Tonga eruption of 15/1/2022.

- Finally, if I understand correctly (please correct me if I am wrong), the observations assimilated here are really $SO_2$ total colum, not volcanic $SO_2$. There are some possibilities to discriminate the volcanic from the non volcanic signal in $SO_2$ total column observations (use of threshold, of flags from the data provider). If the focus of this work is to forecast $SO_2$ from volcanic eruptions, and not from pollution or from fires, some efforts should be carried out to assimilate observations that represent the volcanic signal, not the whole signal.

A bit less important

- Some important aspects of how the data assimilation is carried out (observational error, background error, correlation lengths) are missing or partly missing. Please refer to details in the specific comments.
- Why GOME-2 observations of TC $SO_2$ have not been used for evaluation, or actually for assimilation?

**Specific comments**

- Title: the version of the MOCAGE CTM used for this work should be included in the title
- Lines 1-4: as explained later, aren't the VAACs more interested in ash/sulfate aerosols than $SO_2$? The interest in $SO_2$ is mainly as the precursor of sulfate particles, no?
- Lines 49-57, IFS-COMPO (IFS for atmospheric composition) is used for the MACC/CAMS projects. The "MACC" and "CAMS" systems are really IFS-COMPO. TCSO2 from TROPOMI is operationally assimilated in CAMS since October 2020. Please rephrase this part as I find it confusing
- Line 58 "large amount": are there any quantitative evaluation of the amount of SO2 released?
- Lines 123 – please provide some justifications as to why these flags only have been included. Any study of the impact of using only one/two flags?
- Lines 128-138: why was the layer height product not used to adjust background error statistics and thus add increments preferentially at the "right" altitude?
- Line 163 : what is "accident mode"?
- Line 192 : please detail "many vertical levels"
- Lines 195-198 : what exactly is assimilated? I suppose TC O3 and AOD? Please detail a bit
- Line 230-231 : "For TROPOMI, observation error covariance is directly computed from satellite data" : how is this done? And what is the result?
- Lines 230-235 : it would be good to have a plot that compares the observational error variance for the case study
- Lines 230-235 : what correlation length have been used for the obs covariance matrix?
- Lines 236-240 : what values have the background error variance as compared to the obs error variances for the two sensors?
- Line 246: I think this sentence should come at the end – backed by the conclusion reached before.
- Lines 310-313 : There are other aerosol observations to compare against than MODIS AOD : vertical profiles of extinction from CALIPSO, OMPS-LP etc…

- Line 315 : this evaluation approach is unusual and very nice to see. But it should maybe be restricted to observations that have not been assimilated. If you use assimilated observations to evaluate your analysis, then this is not independent validation.

---

## Author Comment (AC1)

**Answers to the reviewer comments for the paper: "Assimilation of volcanic sulfur dioxide products from IASI and TROPOMI into the chemical transport model MOCAGE: case study of the 2021 La Soufrière Saint-Vincent eruption"**

In the remainder of this document, we provide our full response (in blue) to the reviewers' comments (in black), which details what we did to address their concerns. We include in this document (in italics) some original passages of the article (in green), with modifications and improvements (in red). The following nomenclature has been adopted for the figures and tables: Fig.X are the figures in the initial version of the article, FigRev.x/TabRev.x are the figures/tables in the new version of the article and FigRep.x the plots used exclusively in this document.

**1 General comments**

This new study focuses on the assimilation of volcanic sulphur dioxide ($SO_2$) data from the TROPOMI and IASI satellite instruments into the MOCAGE chemical transport model, using the 2021 La Soufrière Saint-Vincent eruption as a case study. The research highlights the importance of integrating data from different satellite sensors, exploiting their complementary capabilities, to improve real-time atmospheric monitoring and forecasting of volcanic $SO_2$ plumes. The assimilation of combined observations significantly improves the accuracy of $SO_2$ plume predictions compared to using individual data sets, and also captures secondary transformations such as the conversion of $SO_2$ to sulphate aerosols. The study highlights the potential benefits of similar multi-sensor approaches for volcanic hazard monitoring and operational aviation safety.

The study is scientifically sound. The draft paper is well written and mostly clear and concise. I would like to recommend the paper for publication in AMT, subject to the following specific comments and technical corrections given below.

Thank you for your positive evaluation of our study and for your constructive feedback. We are pleased that you find our work scientifically sound, well-written, and clear. The manuscript has been improved as a result and we are grateful for your contributions.

**2 Specific comments**

l204: "The background error covariance is spread on many vertical levels and on many meshgrids thanks to the correlation matrix." Please revisit the sentence and try to be a bit more specific, e.g. what are the actual correlation lengths imposed in the correlation matrix?

We added a paragraph in the article which describes the correlation matrix and gives information about values we used in our study.

*$\mathcal{H}$ is the observation operator used to obtain the model data in the observation space. Before running an assimilation experiment, a full description of R and B matrices is required. The background error covariance is spread in space thanks to the correlation matrix described in [El +20]. This matrix contains both horizontal and vertical components.*

*The horizontal correlation $C^h_{m,n}$ between two points m and n is defined as follows:*

$$C^h_{m,n} = exp[\frac{-d}{2(L_x + L_y)}] \tag{1}$$

where $d$ is the distance between the points $m$ and $n$, $L_x$ and $L_y$ are the longitude and latitude length scales in kilometers. In our study, the longitude and latitude length scales are equal to one meshgrid ($1°$).

In kilometers, length scales become:

$$L_x = L_y = 2R_e.sin(\frac{\pi}{360}) \tag{2}$$

$R_e$ is the Earth's radius (6371.22 km).

The vertical correlation $C^v_{i,j}$ between two pressure levels ($p_i$ and $p_j$) is defined as follows:

$$C^v_{i,j} = exp[-100.[log(\frac{p_i}{p_j})]^2] \tag{3}$$

In our study, the values of the vertical correlation between two consecutive levels are set to 1.

l359: I was wondering why the authors used the Probability of Detection (POD) but not the False Alarm Rate (FAR) and the Critical Success Index (CSI) to evaluate the model results? The POD is a useful metric, but it has limitations, for example if the model is too dispersive and largely overestimates the $SO_2$ plume concentrations, it will produce many hits and a high POD, but overall may not have good predictive quality. Could you please comment on this? Or even better, try to add FAR and CSI estimates?

Yes, we agree POD has limitations, in particular to see if the model is too dispersive. We added in the article the CSI and FAR computed against OMI observations for analyses and forecasts. Nevertheless, we kept a figure with the POD computed when one instrument is assimilated (Fig.6/FigRev.7). It is important to notice that we found a bug in our script used to compute POD. We fixed it and changed the impacted figures in the article in Fig.6/FigRev.7 on the first line of the FigRev.8. and on the first line of the FigRev.11. CSI and FAR metrics are plotted on FigRev.8 and FigRev.11.

**In the "Impact of the assimilation on the detection of $SO_2$ threshold exceedances" section:**

To assess the accuracy of the model in simulating $SO_2$ total columns, a threshold-based analysis was implemented. The goal was to determine the number of instances where both the observations and the model successfully identified $SO_2$ total columns above certain thresholds (labelled as Hits), the instances where the observations exceeded these thresholds but the model failed to detect them (labelled as Misses), the instances where the model exceeds these thresholds but the observations do not reach these thresholds (labelled FalseAlarms) as well as the instances where both the observations and the model successfully identified $SO_2$ total columns under certain thresholds (labelled as CorrectRejections). Using these numbers, we defined three metrics.

The first one is the Probability of Detection (POD), a ratio that ranges from 0 to 1. The POD is calculated by dividing the number of Hits by the sum of Hits and Misses for a given threshold. A POD score of 1 indicates a perfect detection by the model, meaning that all observed instances above the threshold were correctly simulated. On the other hand, a POD of 0 signifies that none of the observed $SO_2$ total columns above the threshold were detected by the model. The POD is computed with the following equation:

$$POD = \frac{Hits}{Hits + Misses} \tag{4}$$

The second one is the Critical Success Index (CSI), a ratio that ranges from 0 to 1. The CSI is calculated by dividing the number of Hits by the sum of Hits, Misses and False Alarms for a given threshold. A CSI score of 1 indicates a perfect detection by the model, meaning that all observed instances above the threshold were correctly simulated. On the other hand, a POD of 0 signifies that

*none of the observed $SO_2$ total columns above the threshold were detected by the model. The CSI is computed with the following equation:*

$$CSI = \frac{Hits}{Hits + Misses + FalseAlarms} \qquad (5)$$

*The last one is the False Alarm Rate (FAR), a ratio that ranges from 0 to 1. The FAR is calculated by dividing the number of False Alarms by the sum of False Alarms and Correct Rejections for a given threshold. A FAR score of 0 indicates that there is only Correct Rejection instances. On the contrary, a FAR score of 1 indicates that there is only False Alarm instances. The FAR is calculated with the following equation:*

$$FAR = \frac{CorrectRejections}{CorrectRejections + FalseAlarms} \qquad (6)$$

*To study these metrics, the notations in the table TabRev.2 are adopted. For POD metrics, times when there are no hits nor misses events are shown by a dot and times when there is no hits bur misses event are represented by a cross. For CSI metrics, times when there is no hits, no misses and no false alarm are shown by a dot, times when there are no hits and no false alarm but misses events occur are represented by a cross. If there is no hits event but if there are misses and false alarms events, a start is plotted. For FAR metrics, a dot is plotted when there are no false alarm and no correct rejections events. A cross is plotted when there are no false alarm event but correct rejections events.*

| | POD | | CSI | | | FAR | |
|---|---|---|---|---|---|---|---|
| | Hits | Misses | Hits | Misses | FalseAlarms | FalseAlarms | CorrectRejections |
| ● | 0 | 0 | 0 | 0 | 0 | 0 | 0 |
| ✕ | 0 | $> 0$ | 0 | $> 0$ | 0 | 0 | $> 0$ |
| ★ | | | 0 | $> 0$ | $> 0$ | | |

TabRev.2: Symbols used in plots according the studied metric and the number of hits, misses, false alarms and correct rejection events.

*Fig.6/FigRev.7 shows the Probability of Detection computed for 1 DU and 5 DU thresholds against TROPOMI and IASI observations. Dots represent times when there is no observation. Crosses represent the moments when simulated $SO_2$ total columns are under a threshold whereas some observations exceeds this threshold. Against TROPOMI instrument, POD values are generally better in the experiments in which TROPOMI observations have been assimilated. In these experiments, POD values exceed 0.75. The POD values are over 0.75 until $12^{th}$ April in iasi_assim experiment except on $9^{th}$ April. POD values decrease at the end of the study period. For the 5 DU threshold, POD values are slightly higher in joint_assim experiment, especially on $10^{th}$ and $11^{th}$ April, when around 100 TROPOMI observations exceed 5 DU. On $12^{th}$ April, POD values are around 0.4 in the experiments in which TROPOMI instrument is assimilated. No $SO_2$ total column higher than 5 DU is simulated for this date in iasi_assim experiment. Between the $13^{th}$ and the $15^{th}$ April and on $9^{th}$ April, no $SO_2$ total column above 5 DU is simulated in MOCAGE. For these days, between 1 and 9 observations above 5 DU are measured by TROPOMI.*

*POD values, computed for a 1 DU threshold and with IASI observations, exceed 0.75 in experiments in which IASI instruments are assimilated. No $SO_2$ total column is simulated with the TROPOMI assimilation until $10^{th}$ April because TROPOMI overpasses the plume after IASI. In the morning of $9^{th}$ April, no observation above 1 DU is detected by IASI. From $11^{th}$ April, POD values vary between 0.3 and 0.8 in the tropomi_assim experiment. In this experiment, POD values are often higher in the morning. In the afternoon of $9^{th}$ April, only one observation above 5 DU is measured by IASI. At this location, the total column is under 5 DU. For this threshold and compared to tropomi_assim experiment, the probability to simulate high $SO_2$ total columns increases thanks to the IASI assimilation. Despite numerous observations above 5 DU, many events are missed on $10^{th}$ April with a POD reaching nearly 0.4 in the morning and 0.5 in the afternoon. Most of simulated $SO_2$ total columns are between 1 DU and 5 DU. The maximum of POD is obtained on $11^{th}$ April after the assimilation of many observations measured by IASI exceeding 5 DU on $10^{th}$ and on $11^{th}$ April.*

[Figure]

Fig.6/FigRev.7: Probability of detection for 1 and 5 DU thresholds for the three experiments: tropomi_assim in blue, iasi_assim in red and joint_assim in green. Dots represent times when there is no observation. Crosses represent the moments when there are only misses events.

[Figure]

FigRev.8: Probability of detection (first line), Critical Success Index (second line) and False Alarm Rate (last line) for 1 and 5 DU thresholds for the three experiments: tropomi_assim in blue, iasi_assim in red and joint_assim in green. The meaning of the symbols is described in the Table ??.

*The first line of the FigRev.8 shows POD computed against OMI observations for 1 DU and 5 DU thresholds. On this line, dots represent times when there are no hits and no misses events. Crosses represent the moments when simulated $SO_2$ total columns are under a threshold whereas some observations exceed this threshold. The POD values computed with a 1 DU threshold are often consistent between the tropomi_assim and the joint_assim experiments. POD values are slightly better in joint_assim experiment between $9^{th}$ to $13^{th}$ April. For the 5 DU threshold, POD value is greater with the joint_assim experiment on $10^{th}$ and on $11^{th}$ April. On $12^{th}$ and $13^{th}$ April, no $SO_2$ total column above 5 DU is modelled by MOCAGE whereas OMI measured observations above this threshold. Elsewhere in the study period, no observation greater than 5 DU is measured by OMI instrument and modelled by MOCAGE.*

*The second line of the FigRev.8 shows CSI computed against OMI observations for 1 DU and 5 DU thresholds. As for POD, the CSI values computed with a 1 DU threshold are often consistent between the tropomi_assim and the joint_assim experiments. CSI values are slightly better in joint_assim experiment on $9^{th}$, on $11^{th}$ and on $12^{th}$ April. On $10^{th}$ and on $11^{th}$, CSI values are around 0.75 whereas POD values are around 0.9. It means that there are few false alarms events during this both days. On the contrary, from the $13^{th}$ April, CSI values are much lower than POD values. It means that the plume in MOCAGE becomes too large, leading to a lot of false alarms events. For the 5 DU threshold, CSI values are better in the joint_assim experiment on $10^{th}$ April, and to a lesser extent on $11^{th}$ April. From $14^{th}$ April, no observations higher than 5 DU are measured by OMI instruments and modelled by MOCAGE. On $9^{th}$, on $12^{th}$ with tropomi_assim experiment and on $13^{th}$ April, no observations greater than 5 DU and observed by OMI instrument but some values above 5 DU are simulated by MOCAGE. On $12^{th}$ April, there are misses and false alarms events in iasi_assim and joint_assim experiments.*

*The third line of the Fig. FigRev.8 shows FAR computed against OMI observations for 1 DU and 5 DU thresholds. The FAR values computed with a 1 DU threshold are similar between experiments. Up to $11^{th}$ April, FAR values are approximatively equal to 0. This shows that the number of correct rejections events is larger than the number of false alarms events. The FAR values increases from $12^{th}$ April meaning that the plume is too large in the model. The FAR values computed with the 5 DU threshold are always close to 0. On $9^{th}$, on $12^{th}$ in tropomi_assim experiment and from $13^{th}$ April, there are only correct rejections events.*

**In the "Impact of assimilation on forecasts" section:**

*Generally speaking, the number of meshes exceeding 1 DU or 5 DU decreases with the forecast period. This was not observed on $9^{th}$ and $10^{th}$ April, when no mesh exceeded 1 DU. The D0, D1 and D2 forecasts show the presence of a plume from $11^{th}$, $12^{th}$ and $13^{th}$ April 2021. These forecasts are initialised by the output of the assimilation of $9^{th}$ April, i.e. before the beginning of the eruption. On $11^{th}$ April, there were around 200 grid cells where the model exceeded 1 DU for the D0 forecast. This number corresponds to the minimum number of grid cells where the TROPOMI and OMI observations reach 1 DU and is below the number of grid cells where the median of the IASI observations reaches 1 DU. On $12^{th}$ April, the plume forecast with D0 was larger, with around 500 meshes exceeding 1 DU, corresponding to the number of meshes where the $25^{th}$ quantile of the TROPOMI observations reached 1 DU and also where the median of the OMI observations reached this threshold. After this date, the number of occurrences of the total column in $SO_2$ exceeding 1 DU increases and becomes greater than the number of grid cells where the IASI and OMI observations reach 1 DU. However, this number remains smaller than the maximum number of meshes calculated using TROPOMI observations. The number of points where the total column reaches 1 DU decreases with the forecast term range. Nevertheless, this number always exceeds the number of grid cells where the OMI and IASI observations reach 1 DU from 14 April onwards. Regarding the 5 DU threshold, no grid cell exceeds this threshold for the D2 forecast. The D1 forecast shows a low number on $12^{th}$ April when the total columns reach 5 DU. However, this number is similar to the number of grid cells where the median of TROPOMI and IASI observations reaches 5 DU. In addition, for this day, the number of points where the model reaches 5 DU is similar between the D0 and D1 forecasts. For $11^{th}$ April, the number of occurrences of a total column greater than or equal to 5 DU is low in the model compared with the UV instruments. This number is within the range of grid cells where IASI observations exceed 5 DU.*

*FigRev.11 represents the POD (on the first line), the CSI (on the second line) and the FAR (on*

*the third line) metrics calculated by comparing the observations measured by OMI and the forecasts at several time steps for 1 DU and 5 DU thresholds. The first forecast is launched on $9^{th}$ April. Consequently, there is no D1 forecast available for this day and no D2 forecast available on on $9^{th}$ and on $10^{th}$ April. The last forecast is launched on $15^{th}$ April so there is no D0 forecast available on $16^{th}$ and on $17^{th}$ April. There is no D1 forecast available on $17^{th}$ April. For POD and CSI metrics computed for the 1 DU threshold, best values are found in the D0 forecast except on $14^{th}$ and on $15^{th}$ April for CSi metric. On $14^{th}$ April CSI are similar between D0 and D1 forecast and on $15^{th}$ April, the CSI is slightly better in D2 forecast. Compared to POD values, CSI values are much lower especially with D0 forecasts on $13^{th}$ and on $14^{th}$ April. It means that there are a lot of location where MOCAGE wrongly simulated total columns stronger than 1 DU. This finding is strengthened by the presence of highest FAR values during these days. POD and CSI metrics computed with a 5 DU threshold show similar values. These metrics are 0 except on $11^{th}$ April with values around 0.2 for both POD and CSI metrics. On $12^{th}$ April both misses and false alarms events occur with the D0 and D1 forecasts. Elsewhere, no $SO_2$ total columns stronger than 5 DU are modelled and no observations stronger than this threshold are observed by the OMI instrument on $14^{th}$, on $15^{th}$ and on $17^{th}$ April. With the 5 Du threshold, there are only correct rejections events.*

[Figure]

FigRev.11: POD on the first line, CSI on the second line and FAR on the last line for 1 DU and 5 DU thresholds for the D0 forecast in blue, the D1 forecast in red and the D2 forecast in green. The meaning of the symbols is described in the Tabrev.2.

l486: It would be good if the text was a bit more specific about how the background error covariances are defined.

To choose the background error covariances, we investigated the values of the background error covariances used in IFS to assimilate $SO_2$. Before 2022, background error standard deviation was defined as a profile containing a peak at 550 hPa, with a valaue of 5e-7 ppv. We decided to lower this specific value for the background error covariances in order to reduce the weight given to the observations.

l491: I'd like to suggest adding a few references regarding the limitations of the chemical modelling and the uncertainties of the meteorological data for the volcanic $SO_2$ chemistry-transport simulations, as these issues have been addressed in several previous studies.

We agree and now cite [WT22] for the uncertainties of the meteorological data and [**schumann2011airborne**] for the limitations of the chemical modelling of volcanic $SO_2$.

l495: Using the information on where the satellites did not actually detect $SO_2$ sounds very helpful, especially to reduce false alarms. In this context it would be good to know if the FAR of the MOCAGE simulations is significant.

We compute FAR for analysis outputs. In our case study, FAR does not reach 0.1 using the TROPOMI instrument. Nevertheless, we observed a slight rise from $12^{th}$ April. However, we computed FAR between 90°W and 40°E and between 20°S and 30°N. Nevertheless, a weak value of FAR means that the number of correct rejections events is very large compared to the number of false alarm events. With TROPOMI, there is a lot of observations. A FAR of 0.1 means that there are around 100 000 false alarms events. So, our model is definitively too dispersive. The use of observations where no $SO_2$ is detected would be useful to correct the size of the plume but it is a real challenge to take that into account during the assimilation, not to mention the data volume that would generate. The 3D-Var technique may also be a limitation in that case.

**3   Technical corrections**

l195: please combine multiple citations in a single set of parentheses
l201: "searched as a sum" $\rightarrow$ "found as a sum"
l215: "15km of high" $\rightarrow$ "15 km of altitude
l226: please use "AVK" or "Avk" consistently
l310 (and other places): please use abbreviations, "figure 4" $\rightarrow$ "Fig. 4" (see AMT manuscript composition guidelines)
l372: please correct "The model did not ??? $SO_2$ total columns..."
l461: "TROPOMI and IASI instruments" $\rightarrow$ "TROPOMI and IASI data"
l463: "more important amount of $SO_2$" $\rightarrow$ do you mean "more realistic"?
l475: "The more the forecast range term is small, the more the plume size is important". $\rightarrow$ Revise/improve sentence?

All technical corrections are taken into account. Moreover, we combined all multiple citations in a single set of parentheses. l463: Yes we do mean realistic. l475: As the forecast period increases, the size of the plume decreases.

**References**

[El +20]   Laaziz El Amraoui et al. "Aerosol data assimilation in the MOCAGE chemical transport model during the TRAQA/ChArMEx campaign: lidar observations". In: *Atmospheric Measurement Techniques* 13.9 (2020), pp. 4645–4667.

[WT22]   Helen N Webster and David J Thomson. "Using ensemble meteorological data sets to treat meteorological uncertainties in a Bayesian volcanic ash inverse modeling system: A case study, Grímsvötn 2011". In: *Journal of Geophysical Research: Atmospheres* 127.24 (2022), e2022JD036469.

---

## Author Comment (AC2)

**Answers to the reviewer comments for the paper: "Assimilation of volcanic sulfur dioxide products from IASI and TROPOMI into the chemical transport model MOCAGE: case study of the 2021 La Soufrière Saint-Vincent eruption"**

In the remainder of this document, we provide our full response (in blue) to the reviewers' comments (in black), which details what we did to address their concerns. We include in this document (in italics) some original passages of the article (in green), with modifications and improvements (in red). The following nomenclature has been adopted for the figures and tables: Fig.X are the figures in the initial version of the article, FigRev.x/TabRev.x are the figures/tables in the new version of the article and FigRep.x the plots used exclusively in this document.

**1 Major comments**

In this article, the authors assimilate observations of $SO_2$ total column from two different sensors (TROPOMI and IASI) into the MOCAGE CTM, in order to provide forecasts of volcanic plume for the VAACs. The topic is of importance, and the paper is well written and easy to read. The authors show a good command of the subject, and the article show promise, but I think it can be improved further. I have a few important questions, which amounts to a major revision. These key points are, in my opinion:

Thank you for your positive evaluation of our study and for your constructive feedback. We are pleased that you find our work well-written, and clear. The manuscript has been improved as a result and we are grateful for your contributions.

The system described here has been designed to work only for this eruption, as mentioned by the authors themselves in the conclusion, because of the setting of the background error standard deviation (lines 239-240). If applied, say, to the Pinatubo eruption, adding increments between 9 and 21 km altitude will probably give a very wrong profile. Isn't it possible to design a system that doesn't need the manual input of this key information about injection height? Ideally, aerosol layer height information should be used.

Indeed, these settings only work on this eruption. For Pinatubo, the background error profile would have to be changed to add $SO_2$ at the true altitude. Care must be taken when using aerosol heights. During eruptions, $SO_2$ is emitted as well as ash. Ash and $SO_2$ emissions are not always at the same altitudes ([Mox+14]). In this case, using aerosols can also lead to a false distribution and even a plume with an incorrect shape. Another way is too use information about the height altitude of the $SO_2$ plume but it is only available when observations reach 20 DU for TROPOMI_LH. We had observations with altitudes on $10^{th}$ and on $11^{th}$ April 2021 with TROPOMI_LH. We added a sentence indicating that this setting is only valid for this eruption in the 'TROPOMI and IASI data assimilation setup' section.

The main objective of the MOCAGE CTM in the VAACs is to provide forecasts of the aerosol layer (ash or sulphuric acid). I understand that ash in not in the scope of this article. However, the aerosol validation aspect is treated very shortly, as compared to the $SO_2$ aspect. The volcanic signal is small as compared to the tropospheric aerosol signal for this eruption, which complicates things. The authors should try to either find and compare against retrievals which clearly show a volcanic signal for this eruption (OMPS-LP possibly), or chose another eruption for which the aerosol signal has been

extensively documented, such as the Hunga Tonga eruption of 15/1/2022.

The main objectives of the VAAC is to provide forecasts of the ash and $SO_2$. This is why we have not focused on non-ash aerosols. We want to conduct the assimilation of $SO_2$ for common eruptions like La Soufière Saint-Vincent eruption. Other more recent eruptions (eg. Popocatepetl) were more prone to sulfate aerosol production, leading to a clearer signal in areas where no other types of aerosols were abundant. Unfortunately, as stated in the submitted article, sulfate aerosols were not easily identified in our case study. The case of the Hunga Tonga would be interesting to study but it was an exceptional event.

Finally, if I understand correctly (please correct me if I am wrong), the observations assimilated here are really $SO_2$ total column, not volcanic $SO_2$. There are some possibilities to discriminate the volcanic from the non volcanic signal in $SO_2$ total column observations (use of threshold, of flags from the data provider). If the focus of this work is to forecast $SO_2$ from volcanic eruptions, and not from pollution or from fires, some efforts should be carried out to assimilate observations that represent the volcanic signal, not the whole signal.

In fact, this is a total $SO_2$ column, with a volcanic component and an anthropogenic component. The flags do not help us to differentiate between anthropogenic and volcanic $SO_2$. TROPOMI provides flags for each observation: 0 when there is no $SO_2$, 1 when there is $SO_2$, 2 when there is $SO_2$ near a known volcano and 3 when there is $SO_2$ near a known anthropogenic area. This flag is always 2 up to 300 km from the volcano , and become 1 when the distance from the volcano is bigger than a value. Consequently, when $SO_2$ is transported, this flag becomes 1 or 3 if the volcanic $SO_2$ is transported less than 100 km from an anthropogenic zone. The use of this flag therefore makes it impossible to differentiate between volcanic and anthropogenic $SO_2$. The use of a threshold is an excellent idea, but we did not apply it in this case because in the days before the eruption, no $SO_2$ was detected in this area. On the other hand, in the operational version of TROPOMI $SO_2$ assimilation, we only assimilate observations greater than 3DU over the global domain. This avoids falsely considering volcanic $SO_2$ in the Norilsk region, for instance.

Some important aspects of how the data assimilation is carried out (observational error, background error, correlation lengths) are missing or partly missing. Please refer to details in the specific comments.

We added a paragraph in the article which describes the correlation matrix and gives information about values we used in our study.

*$\mathcal{H}$ is the observation operator used to obtain the model data in the observation space. Before running an assimilation experiment, a full description of R and B matrices is required. The background error covariance is spread in space thanks to the correlation matrix described in [El +20]. This matrix contains both horizontal and vertical components.*

*The horizontal correlation $C_{m,n}^h$ between two points m and n is defined as follows:*

$$C_{m,n}^h = exp[\frac{-d}{2(L_x + L_y)}] \tag{1}$$

*where d is the distance between the points m and n, $L_x$ and $L_y$ are the longitude and latitude length scales in kilometers. In our study, the longitude and latitude length scales are equal to one meshgrid (1°).*
*In kilometers, length scales become:*

$$L_x = L_y = 2R_e.sin(\frac{\pi}{360}) \tag{2}$$

*$R_e$ is the Earth's radius (6371.22 km).*
*The vertical correlation $C_{i,j}^v$ between two pressure levels ($p_i$ and $p_j$) is defined as follows:*

$$C_{i,j}^v = exp[-100.[log(\frac{p_i}{p_j})]^2] \tag{3}$$

*In our study, the values of the vertical correlation between two consecutive levels are set to 1.*

Why GOME-2 observations of TC SO$_2$ have not been used for evaluation, or actually for assimilation?

GOME-2 SO$_2$ products are known to be noisier than other products. Moreover, GOME-2 and IASI instruments are on the same satellite. Consequently, the assimilation of these instruments would take place at the same time which can be a problem is the information given by instruments are contradictory. We choose to assimilate IASI because it is an IR instruments allowing to have information during the night.

**2   Specific comments**

Title: the version of the MOCAGE CTM used for this work should be included in the title.

We used the MOCAGE version of March 2022. We added this to the main title of the article.

Lines 1-4: as explained later, aren't the VAACs more interested in ash/sulfate aerosols than SO$_2$? The interest in SO$_2$ is mainly as the precursor of sulfate particles, no?

The VAAC's mission is to forecast volcanic ash as well as volcanic SO$_2$, mainly for the impact of this gaseous compound on health and smell within the aircrafts. There is no request on sulfate aerosols.

Lines 49-57, IFS-COMPO (IFS for atmospheric composition) is used for the MACC/CAMS projects. The "MACC" and "CAMS" systems are really IFS-COMPO. TCSO$_2$ from TROPOMI is operationally assimilated in CAMS since October 2020. Please rephrase this part as I find it confusing.

Thank you for the precision. We rephrased this part.

*Assimilation of volcanic sulfur dioxide observations into a model has already been performed in several situations like for the eruption events of the Eyjafjallajökull in 2010 and the Grímsvötn in 2011. Volcanic sulfur dioxide released by these volcanoes has been monitored by the GOME-2 (Global Ozone Monitoring Experiment-2) and OMI (Ozone Monitoring Instrument) UV sensors. The resulting retrieved observations have been assimilated in the Integrated Forecasting System for atmospheric composition (IFS-COMPO) of the European Centre for Medium-Range Weather Forecasts (ECMWF) and improved the sulfur dioxide plume forecasts [FI13]. Volcanic sulfur dioxide retrievals from GOME-2 and TROPOMI (Tropospheric Monitoring Instrument) UV sensors observations are operationally assimilated in the global IFS-COMPO assimilation system since October 2020. On top of that, a retrieved volcanic sulfur dioxide layer height from the TROPOMI Layer Height product has been assimilated.*

Line 58 "large amount": are there any quantitative evaluation of the amount of SO$_2$ released?

Around 380 kt of SO$_2$ are released during the 2 first days of the eruption ([Ess+24]). We did not find any study on the amount of SO$_2$ emitted during the entire eruption. We added this precision in the paper.

Lines 123 – please provide some justifications as to why these flags only have been included. Any study of the impact of using only one/two flags?

We tested assimilation using different flag values. When we used flag 1, the majority of non-zero observations were assimilated. The plume is consistent with that observed from other instruments. However, if we take the FigRep.1 as an example, observations less than 300 km from the volcano are not assimilated in this case. On the contrary, if only observations with a flag of 2 are assimilated, only observations within 300 km of the volcano are assimilated. When SO$_2$ is transported in the model and is located more than 300 km from the volcano, the plume is not corrected in the model because there are no observations with a flag of 2 more than 300 km from the volcano. We have also taken flag 3

into account because it can happen that the SO$_2$ emitted by a volcano and then transported comes close (within 100km) to an area known for its SO$_2$ emissions. In our case study, the observations with a flag of 3 are few compared to the other flags (FigRep.2) and do not have a significant impact on the assimilation outputs. Although information on pixels where SO$_2$ has not been detected (flag 0) could be useful for correcting the shape of the plume, these observations have not been taken into account but will be the subject of a future study. However, many questions remain concerning the method to be used to assimilate them, as well as the volume of data. We did not take flag 4 because the documentation announces the possibility of false-positive detection.

[Figure]

FigRep.1: Values of detection flag on 10$^{th}$ April 2021 at 17 UTC.

[Figure]

FigRep.2: Number of observations by detection flag values between 9$^{th}$ and 15$^{th}$ April 2021.

Lines 128-138: why was the layer height product not used to adjust background error statistics and thus add increments preferentially at the "right" altitude?

This comment is very relevant and this topic will be the subject of a forthcoming publication. Nevertheless, some difficulties may be encountered when we consider the altitude of the plume. First, the altitude of the $SO_2$ plume is only available for total columns stronger than 20 DU for TROPOMI. In our case, there are observations matching this criteria on $10^{th}$ and $11^{th}$ April. Moreover, due to the fine resolution of TROPOMI, the distribution of the altitude diagnosed by the layer height product can be very large within a mesh. When it happens, the simulated plume can be thick. IASI has the advantage to provide information about the height of the plume whatever the observations. However, there is no information about the plume thickness with IASI. Consequently, we need to make hypothesis about the thickness of the plume. With TROPOMI, the plume has a 2.5 km thickness.

Line 163 : what is "accident mode"?

The accident mode is a version of MOCAGE allowing to study the transport of a punctual emission fastly because there is no chemical reactions. It is used for industrial/nuclear accident or during a volcanic eruption to forecast the dispersion of the relased coumponents. We reformulated the sentence.

Line 192 : please detail "many vertical levels"

We realised that the sentence was badly formulated: "at different vertical levels" instead of "at many vertical levels" would have been more appropriate. The biomass burning emissions are injected at different altitudes according to the latitude of the wildfires, at 1km of altitude in the tropics, at 2 km of altitude in the middle latitudes and at 6 km of altitude in the high latitudes. We added altitudes of the emissions in the article.

Lines 195-198 : what exactly is assimilated? I suppose TC O3 and AOD? Please detail a bit

We described a bit more the products that can be assimilated in the paper.

Many products can be assimilated in MOCAGE. For example, to improve $O_3$ in MOCAGE, total column [Emi+14] or radiances [EEG21; VGF24] can be assimilated. For the aerosols, AOD [Sič+16; El +22] or lidar observations [El +20; Cor+23] can be assimilated in MOCAGE.

The assimilation system used in this study is the 3D-VAR, described hereafter. A short-range forecast from MOCAGE $x^b$ and observations $y$ are combined to find the optimal state $x^a$, taking into account their respective error covariance matrices B and R. $x^a$ can be found as the sum of $x^b + \delta x^a$ where $\delta x^a$ is the increment minimising the cost function J:

Line 230-231 : "For TROPOMI, observation error covariance is directly computed from satellite data" : how is this done? And what is the result?

Thank you for the comment, this sentence was badly formulated. TROPOMI instrument provides an information about the uncertainties $(U_n)$ of the observations. We used this information to compute the observation error covariance (R) as follows:

$$R = U_n^2 \tag{4}$$

We rephrased this part.

In this study, we assume there is no spatial correlation in the observation error. For TROPOMI, observation error standard deviation (sigR) is set according to the uncertainties provided by the instrument for each observation (FigRev.1 on the left). The observations error standard deviation is set to around 25% of the observations for TROPOMI. The uncertainties of IASI measurements vary according to the value of the total column measured. In the case of this eruption, IASI measured total $SO_2$ columns ranging from 0.5 to 20 DU. The uncertainties in this range of observations vary from around

*25% to 5% of the observation [Cla+12]. For IASI, we set the observation error standard deviation to 15% of the observation values (FigRev.1 on the right).*

Lines 230-235 : it would be good to have a plot that compares the observational error variance for the case study

Yes, we agree with your comment. We added a plot (FigRev. 1) with the observation error standard deviation that we prescribed according to the observations values. We made two plots: one for the TROPOMI instrument and the other one for IASI instruments.

[Figure]

Figrev.1: Values of the observation error standard deviation (sigR) according to TROPOMI (on the left) and IASI (on the right) observations.

Lines 230-235 : what correlation length have been used for the obs covariance matrix?

The correlation length for the observation covariance matrix is set to 0. This setting is widely chosen even among NWP assimilation systems, mainly because its implementation is not straightforward.

Lines 236-240 : what values have the background error variance as compared to the obs error variances for the two sensors?

This is an interesting question but it is hard to answer. Indeed, the background error standard deviation is a 3D concentration and the observation error standard deviation has the dimension of a total column. To compare this parameters, it would be necessary to apply the averaging kernels to the background error standard deviation to have the same dimensions, which is not so straightforward.

Line 246: I think this sentence should come at the end – backed by the conclusion reached before.

This sentence has been added to the end of the "Impact on $SO_2$ and sulfate" part.

Lines 310-313 : There are other aerosol observations to compare against than MODIS AOD : vertical profiles of extinction from CALIPSO, OMPS-LP etc...

We looked at the vertical profiles of extinction from CALIPSO. Some values are stronger around 9-10 km of altitude. Nevertheless, according to the instrument, this is often clouds. Moreover, for this eruption, plume is under the tropopause. Consequently, no stratospheric aerosol is detected by

CALIPSO.

Line 315 : this evaluation approach is unusual and very nice to see. But it should maybe be restricted to observations that have not been assimilated. If you use assimilated observations to evaluate your analysis, then this is not independent validation.

We plotted the POD against TROPOMI and IASI because these instruments are not assimilated in all experiments. Both instruments are assimilated on the joint_assim experiment. It allows to see the improvement of the POD against TROPOMI when IASI instruments are assimilated and the improvement of POD againt IASI when TROPOMI instrument is assimilated. We want to inform you that the values of POD have been modified on the Fig.6/FigRev.7 because of a bug in one of our scripts. We computed CSI and FAR againt OMI observations to show that MOCAGE is too dispersive. These metrics are plotted on the FigRev.8 for the analysis study and on FigRev.11 for the forecasts study.

**In the "Impact of the assimilation on the detection of SO$_2$ threshold exceedances" section:**

*To assess the accuracy of the model in simulating SO$_2$ total columns, a threshold-based analysis was implemented. The goal was to determine the number of instances where both the observations and the model successfully identified SO$_2$ total columns above certain thresholds (labelled as Hits), the instances where the observations exceeded these thresholds but the model failed to detect them (labelled as Misses), the instances where the model exceeds these thresholds but the observations do not reach these thresholds (labelled FalseAlarms) as well as the instances where both the observations and the model successfully identified SO$_2$ total columns under certain thresholds (labelled as CorrectRejections). Using these numbers, we defined three metrics.*

*The first one is the Probability of Detection (POD), a ratio that ranges from 0 to 1. The POD is calculated by dividing the number of Hits by the sum of Hits and Misses for a given threshold. A POD score of 1 indicates a perfect detection by the model, meaning that all observed instances above the threshold were correctly simulated. On the other hand, a POD of 0 signifies that none of the observed SO$_2$ total columns above the threshold were detected by the model. The POD is computed with the following equation:*

$$POD = \frac{Hits}{Hits + Misses} \tag{5}$$

*The second one is the Critical Success Index (CSI), a ratio that ranges from 0 to 1. The CSI is calculated by dividing the number of Hits by the sum of Hits, Misses and False Alarms for a given threshold. A CSI score of 1 indicates a perfect detection by the model, meaning that all observed instances above the threshold were correctly simulated. On the other hand, a POD of 0 signifies that none of the observed SO$_2$ total columns above the threshold were detected by the model. The CSI is computed with the following equation:*

$$CSI = \frac{Hits}{Hits + Misses + FalseAlarms} \tag{6}$$

*The last one is the False Alarm Rate (FAR), a ratio that ranges from 0 to 1. The FAR is calculated by dividing the number of False Alarms by the sum of False Alarms and Correct Rejections for a given threshold. A FAR score of 0 indicates that there is only Correct Rejection instances. On the contrary, a FAR score of 1 indicates that there is only False Alarm instances. The FAR is calculated with the following equation:*

$$FAR = \frac{CorrectRejections}{CorrectRejections + FalseAlarms} \tag{7}$$

*To study these metrics, the notations in the table TabRev.2 are adopted. For POD metrics, times when there are no hits nor misses events are shown by a dot and times when there is no hits bur misses event are represented by a cross. For CSI metrics, times when there is no hits, no misses and no false alarm are shown by a dot, times when there are no hits and no false alarm but misses events occur are represented by a cross. If there is no hits event but if there are misses and false alarms events, a start*

is plotted. For FAR metrics, a dot is plotted when there are no false alarm and no correct rejections events. A cross is plotted when there are no false alarm event but correct rejections events.

| | POD | | CSI | | | FAR | |
|---|---|---|---|---|---|---|---|
| | Hits | Misses | Hits | Misses | FalseAlarms | FalseAlarms | CorrectRejections |
| ● | 0 | 0 | 0 | 0 | 0 | 0 | 0 |
| × | 0 | $> 0$ | 0 | $> 0$ | 0 | 0 | $> 0$ |
| ★ | | | 0 | $> 0$ | $> 0$ | | |

TabRev.2: Symbols used in plots according the studied metric and the number of hits, misses, false alarms and correct rejection events.

Fig.6/FigRev.7 shows the Probability of Detection computed for 1 DU and 5 DU thresholds against TROPOMI and IASI observations. Dots represent times when there is no observation. Crosses represent the moments when simulated $SO_2$ total columns are under a threshold whereas some observations exceeds this threshold. Against TROPOMI instrument, POD values are generally better in the experiments in which TROPOMI observations have been assimilated. In these experiments, POD values exceed 0.75. The POD values are over 0.75 until $12^{th}$ April in iasi_assim experiment except on $9^{th}$ April. POD values decrease at the end of the study period. For the 5 DU threshold, POD values are slightly higher in joint_assim experiment, especially on $10^{th}$ and $11^{th}$ April, when around 100 TROPOMI observations exceed 5 DU. On $12^{th}$ April, POD values are around 0.4 in the experiments in which TROPOMI instrument is assimilated. No $SO_2$ total column higher than 5 DU is simulated for this date in iasi_assim experiment. Between the $13^{th}$ and the $15^{th}$ April and on $9^{th}$ April, no $SO_2$ total column above 5 DU is simulated in MOCAGE. For these days, between 1 and 9 observations above 5 DU are measured by TROPOMI.

POD values, computed for a 1 DU threshold and with IASI observations, exceed 0.75 in experiments in which IASI instruments are assimilated. No $SO_2$ total column is simulated with the TROPOMI assimilation until $10^{th}$ April because TROPOMI overpasses the plume after IASI. In the morning of $9^{th}$ April, no observation above 1 DU is detected by IASI. From $11^{th}$ April, POD values vary between 0.3 and 0.8 in the tropomi_assim experiment. In this experiment, POD values are often higher in the morning. In the afternoon of $9^{th}$ April, only one observation above 5 DU is measured by IASI. At this location, the total column is under 5 DU. For this threshold and compared to tropomi_assim experiment, the probability to simulate high $SO_2$ total columns increases thanks to the IASI assimilation. Despite numerous observations above 5 DU, many events are missed on $10^{th}$ April with a POD reaching nearly 0.4 in the morning and 0.5 in the afternoon. Most of simulated $SO_2$ total columns are between 1 DU and 5 DU. The maximum of POD is obtained on $11^{th}$ April after the assimilation of many observations measured by IASI exceeding 5 DU on $10^{th}$ and on $11^{th}$ April.

The first line of the FigRev.8 shows POD computed against OMI observations for 1 DU and 5 DU thresholds. On this line, dots represent times when there are no hits and no misses events. Crosses represent the moments when simulated $SO_2$ total columns are under a threshold whereas some observations exceed this threshold. The POD values computed with a 1 DU threshold are often consistent between the tropomi_assim and the joint_assim experiments. POD values are slightly better in joint_assim experiment between $9^{th}$ to $13^{th}$ April. For the 5 DU threshold, POD value is greater with the joint_assim experiment on $10^{th}$ and on $11^{th}$ April. On $12^{th}$ and $13^{th}$ April, no $SO_2$ total column above 5 DU is modelled by MOCAGE whereas OMI measured observations above this threshold. Elsewhere in the study period, no observation greater than 5 DU is measured by OMI instrument and modelled by MOCAGE.

The second line of the FigRev.8 shows CSI computed against OMI observations for 1 DU and 5 DU thresholds. As for POD, the CSI values computed with a 1 DU threshold are often consistent between the tropomi_assim and the joint_assim experiments. CSI values are slightly better in joint_assim experiment on $9^{th}$, on $11^{th}$ and on $12^{th}$ April. On $10^{th}$ and on $11^{th}$, CSI values are around 0.75 whereas POD values are around 0.9. It means that there are few false alarms events during this both days. On the contrary, from the $13^{th}$ April, CSI values are much lower than POD values. It means that the plume in MOCAGE becomes too large, leading to a lot of false alarms events. For the 5 DU threshold, CSI values are better in the joint_assim experiment on $10^{th}$ April, and to a lesser extent on $11^{th}$ April.

[Figure]

Fig.6/FigRev.7: Probability of detection for 1 and 5 DU thresholds for the three experiments: tropomi_assim in blue, iasi_assim in red and joint_assim in green. Dots represent times when there is no observation. Crosses represent the moments when there are only misses events.

[Figure]

FigRev.8: Probability of detection (first line), Critical Success Index (second line) and False Alarm Rate (last line) for 1 and 5 DU thresholds for the three experiments: tropomi_assim in blue, iasi_assim in red and joint_assim in green. The meaning of the symbols is described in the Table ??.

*From 14th April, no observations higher than 5 DU are measured by OMI instruments and modelled by MOCAGE. On 9th, on 12th with tropomi_assim experiment and on 13th April, no observations greater than 5 DU and observed by OMI instrument but some values above 5 DU are simulated by MOCAGE. On 12th April, there are misses and false alarms events in iasi_assim and joint_assim experiments.*

*The third line of the Fig. FigRev.8 shows FAR computed against OMI observations for 1 DU and 5 DU thresholds. The FAR values computed with a 1 DU threshold are similar between experiments. Up to 11th April, FAR values are approximatively equal to 0. This shows that the number of correct rejections events is larger than the number of false alarms events. The FAR values increases from 12th April meaning that the plume is too large in the model. The FAR values computed with the 5 DU threshold are always close to 0. On 9th, on 12th in tropomi_assim experiment and from 13th April, there are only correct rejections events.*

**In the "Impact of assimilation on forecasts" section:**

*Generally speaking, the number of meshes exceeding 1 DU or 5 DU decreases with the forecast period. This was not observed on 9th and 10th April, when no mesh exceeded 1 DU. The D0, D1 and D2 forecasts show the presence of a plume from 11th, 12th and 13th April 2021. These forecasts are initialised by the output of the assimilation of 9th April, i.e. before the beginning of the eruption. On 11th April, there were around 200 grid cells where the model exceeded 1 DU for the D0 forecast. This number corresponds to the minimum number of grid cells where the TROPOMI and OMI observations reach 1 DU and is below the number of grid cells where the median of the IASI observations reaches 1 DU. On 12th April, the plume forecast with D0 was larger, with around 500 meshes exceeding 1 DU, corresponding to the number of meshes where the 25th quantile of the TROPOMI observations reached 1 DU and also where the median of the OMI observations reached this threshold. After this date, the number of occurrences of the total column in $SO_2$ exceeding 1 DU increases and becomes greater than the number of grid cells where the IASI and OMI observations reach 1 DU. However, this number remains smaller than the maximum number of meshes calculated using TROPOMI observations. The number of points where the total column reaches 1 DU decreases with the forecast term range. Nevertheless, this number always exceeds the number of grid cells where the OMI and IASI observations reach 1 DU from 14 April onwards. Regarding the 5 DU threshold, no grid cell exceeds this threshold for the D2 forecast. The D1 forecast shows a low number on 12th April when the total columns reach 5 DU. However, this number is similar to the number of grid cells where the median of TROPOMI and IASI observations reaches 5 DU. In addition, for this day, the number of points where the model reaches 5 DU is similar between the D0 and D1 forecasts. For 11th April, the number of occurrences of a total column greater than or equal to 5 DU is low in the model compared with the UV instruments. This number is within the range of grid cells where IASI observations exceed 5 DU.*

*FigRev.11 represents the POD (on the first line), the CSI (on the second line) and the FAR (on the third line) metrics calculated by comparing the observations measured by OMI and the forecasts at several time steps for 1 DU and 5 DU thresholds. The first forecast is launched on 9th April. Consequently, there is no D1 forecast available for this day and no D2 forecast available on on 9th and on 10th April. The last forecast is launched on 15th April so there is no D0 forecast available on 16th and on 17th April. There is no D1 forecast available on 17th April. For POD and CSI metrics computed for the 1 DU threshold, best values are found in the D0 forecast except on 14th and on 15th April for CSi metric. On 14th April CSI are similar between D0 and D1 forecast and on 15th April, the CSI is slightly better in D2 forecast. Compared to POD values, CSI values are much lower especially with D0 forecasts on 13th and on 14th April. It means that there are a lot of location where MOCAGE wrongly simulated total columns stronger than 1 DU. This finding is strengthened by the presence of highest FAR values during these days. POD and CSI metrics computed with a 5 DU threshold show similar values. These metrics are 0 except on 11th April with values around 0.2 for both POD and CSI metrics. On 12th April both misses and false alarms events occur with the D0 and D1 forecasts. Elsewhere, no $SO_2$ total columns stronger than 5 DU are modelled and no observations stronger than this threshold are observed by the OMI instrument on 14th, on 15th and on 17th April. With the 5 Du threshold, there are only correct rejections events.*

[Figure]

FigRev.11: POD on the first line, CSI on the second line and FAR on the last line for 1 DU and 5 DU thresholds for the D0 forecast in blue, the D1 forecast in red and the D2 forecast in green. The meaning of the symbols is described in the Tabrev.2.

**References**

[Cla+12]    L. Clarisse et al. "Retrieval of sulphur dioxide from the infrared atmospheric sounding interferometer (IASI)". In: *Atmospheric Measurement Techniques* 5.3 (Mar. 13, 2012). Publisher: Copernicus GmbH, pp. 581–594. ISSN: 1867-1381. DOI: 10.5194/amt-5-581-2012. URL: https://amt.copernicus.org/articles/5/581/2012/ (visited on 09/18/2024).

[FI13]      Johannes Flemming and Antje Inness. "Volcanic sulfur dioxide plume forecasts based on UV satellite retrievals for the 2011 Grímsvötn and the 2010 Eyjafjallajökull eruption". In: *Journal of Geophysical Research: Atmospheres* 118.17 (2013). _eprint: https://onlinelibrary.wiley.com/doi/pdf/ pp. 10, 172–10, 189. ISSN: 2169-8996. DOI: 10.1002/jgrd.50753. URL: https://onlinelibrary.wiley.com/doi/abs/10.1002/jgrd.50753 (visited on 09/18/2024).

[Emi+14]    E. Emili et al. "Combined assimilation of IASI and MLS observations to constrain tropospheric and stratospheric ozone in a global chemical transport model". In: *Atmospheric Chemistry and Physics* 14.1 (Jan. 8, 2014). Publisher: Copernicus GmbH, pp. 177–198. ISSN: 1680-7316. DOI: 10.5194/acp-14-177-2014. URL: https://acp.copernicus.org/articles/14/177/2014/ (visited on 09/18/2024).

[Mox+14]    Eldbjørg Dirdal Moxnes et al. "Separation of ash and sulfur dioxide during the 2011 Grímsvötn eruption". In: *Journal of Geophysical Research: Atmospheres* 119.12 (2014), pp. 7477–7501.

[Sič+16]    Bojan Sič et al. "Aerosol data assimilation in the chemical transport model MOCAGE during the TRAQA/ChArMEx campaign: aerosol optical depth". In: *Atmospheric Measurement Techniques* 9.11 (Nov. 22, 2016). Publisher: Copernicus GmbH, pp. 5535–5554. ISSN: 1867-1381. DOI: 10.5194/amt-9-5535-2016. URL: https://amt.copernicus.org/articles/9/5535/2016/ (visited on 09/18/2024).

[El +20]    Laaziz El Amraoui et al. "Aerosol data assimilation in the MOCAGE chemical transport model during the TRAQA/ChArMEx campaign: lidar observations". In: *Atmospheric Measurement Techniques* 13.9 (2020), pp. 4645–4667.

[EEG21]    Mohammad El Aabaribaoune, Emanuele Emili, and Vincent Guidard. "Estimation of the error covariance matrix for IASI radiances and its impact on the assimilation of ozone in a chemistry transport model". In: *Atmospheric Measurement Techniques* 14.4 (Apr. 13, 2021). Publisher: Copernicus GmbH, pp. 2841–2856. ISSN: 1867-1381. DOI: 10.5194/amt-14-2841-2021. URL: https://amt.copernicus.org/articles/14/2841/2021/ (visited on 09/18/2024).

[El +22]    Laaziz El Amraoui et al. "A Pre-Operational System Based on the Assimilation of MODIS Aerosol Optical Depth in the MOCAGE Chemical Transport Model". In: *Remote Sensing* 14.8 (Jan. 2022). Number: 8 Publisher: Multidisciplinary Digital Publishing Institute, p. 1949. ISSN: 2072-4292. DOI: 10.3390/rs14081949. URL: https://www.mdpi.com/2072-4292/14/8/1949 (visited on 09/18/2024).

[Cor+23]    Flavien Cornut et al. "Assimilation of Aerosol Observations from the Future Spaceborne Lidar Onboard the AOS Mission into the MOCAGE Chemistry: Transport Model". In: *Proceedings of the 30th International Laser Radar Conference*. Ed. by John T. Sullivan et al. Cham: Springer International Publishing, 2023, pp. 645–651. ISBN: 978-3-031-37818-8. DOI: 10.1007/978-3-031-37818-8_83.

[Ess+24]    Ben Esse et al. "SO2 emissions during the 2021 eruption of La Soufrière, St Vincent, revealed with back-trajectory analysis of TROPOMI imagery". In: *The 2020–21 Eruption of La Soufrière Volcano, St Vincent*. Geological Society of London, Jan. 2024. ISBN: 9781786205964. DOI: 10.1144/SP539-2022-77. URL: https://doi.org/10.1144/SP539-2022-77.

[VGF24]    Francesca Vittorioso, Vincent Guidard, and Nadia Fourrié. "Assessment of the contribution of the Meteosat Third Generation Infrared Sounder (MTG-IRS) for the characterisation of ozone over Europe". In: *Atmospheric Measurement Techniques* 17.17 (Sept. 12, 2024). Publisher: Copernicus GmbH, pp. 5279–5299. ISSN: 1867-1381. DOI: 10.5194/amt-17-5279-2024. URL: https://amt.copernicus.org/articles/17/5279/2024/ (visited on 09/18/2024).